# SafeGenes: Evaluating the Adversarial Robustness of Genomic Foundation Models

Huixin Zhan[1,*], Clovis Barbour[1,†], and Jason H. Moore[2,*]

[1]Department of Computer Science & Engineering, New Mexico Tech, NM, USA
[2]Computational Biomedicine, Cedars-Sinai Medical Center, CA, USA
[*]Corresponding authors: huixin.zhan@nmt.edu, jason.moore@csmc.edu
[†]Contributing author: clovis.barbour@student.nmt.edu

## SUMMARY

Genomic Foundation Models (GFMs), such as Evolutionary Scale Modeling (ESM), have demonstrated significant success in variant effect prediction. However, their adversarial robustness remains largely unexplored. To address this gap, we propose **SafeGenes**: a framework for Secure analysis of genomic foundation models, leveraging adversarial attacks to evaluate robustness against both engineered near-identical adversarial Genes and embedding-space manipulations. In this study, we assess the adversarial vulnerabilities of GFMs using two approaches: the Fast Gradient Sign Method (FGSM) and a soft prompt attack. FGSM introduces minimal perturbations to input sequences, while the soft prompt attack optimizes continuous embeddings to manipulate model predictions without modifying the input tokens. By combining these techniques, SafeGenes provides a comprehensive assessment of GFM susceptibility to adversarial manipulation. Targeted soft prompt attacks induced severe degradation in MLM-based shallow architectures such as ProteinBERT, while still producing substantial failure modes even in high-capacity foundation models such as ESM1b and ESM1v. These findings expose critical vulnerabilities in current foundation models, opening new research directions toward improving their security and robustness in high-stakes genomic applications such as variant effect prediction.

**The Bigger Picture**

Advances in large language models have transformed the way we analyze language, images, and scientific data—and they are now reshaping biomedicine. Genomic foundation models (GFMs), which apply language modeling principles to DNA and protein sequences, are emerging as powerful tools to predict the functional effects of genetic mutations. These models promise to accelerate the diagnosis of inherited diseases, personalize treatments, and uncover novel insights into human biology. However, with increasing adoption in clinical genomics, the question arises: how secure and trustworthy are these models when faced with unexpected or malicious inputs?

In this study, we present *SafeGenes*, a framework to evaluate the adversarial robustness of GFMs. We show that these models, despite their impressive performance, remain vulnerable to carefully crafted perturbations—either in the form of subtle changes to input sequences or imperceptible manipulations of their internal prompt structure. Our targeted soft prompt attacks can cause confident misclassification of benign variants, revealing latent vulnerabilities that are not captured by traditional robustness tests. These vulnerabilities are particularly concerning in medical contexts, where decisions influenced by model outputs can directly impact patient outcomes.

Beyond demonstrating specific failure cases, our findings have broader implications for the future of trustworthy AI in genomics. Current defense mechanisms, which rely on input-level checks or confidence scores, may not suffice when attacks operate within the semantic space of the model itself. Our work suggests a path forward: integrating robustness evaluation into the development cycle of genomic AI systems and designing future defenses that monitor changes in internal representation space. Just as medical instruments must pass rigorous safety tests, genomic models must be stress-tested for reliability before clinical use. *SafeGenes* is a step toward that future.

## INTRODUCTION

Genomic Foundation Models (GFMs), such as Evolutionary Scale Modeling (ESM), have revolutionized the field of genomics by providing powerful tools for variant effect prediction and genomic sequence analysis. These models leverage large-scale data and sophisticated architectures to deliver high accuracy and generalizability across diverse tasks, offering transformative potential in biomedical applications. For instance, AlphaMissense [1] integrates evolutionary conservation and structural modeling for clinical variant classification, while ESM1b [2] demonstrates strong zero-shot performance on isoform-specific missense predictions using large-scale masked language modeling. Despite these advances, current GFMs are generally trained in a task-agnostic manner and lack disease-specific adaptation, limiting their direct applicability in clinical decision-making. Moreover, their deployment in critical domains such as clinical diagnostics necessitates rigorous evaluation of their reliability under adversarial conditions. In particular, it remains unclear whether GFMs are vulnerable to adversarial manipulations—whether through perturbed input sequences or learned prompts—which could compromise model robustness and trustworthiness.

To explore this question in a clinically grounded setting, we fine-tune a range of GFMs for disease-specific variant effect prediction (VEP) using DYNA [3]—a modular framework that

adapts protein- and DNA-based GFMs to cardiac and regulatory genomics tasks. DYNA employs a Siamese neural network architecture (Figure 1a) and a pseudo-log-likelihood ratio (PLLR)-based scoring function (Figure 1b), enabling effective adaptation to domain-specific VEP with limited rare variant data. Using this framework, we fine-tune multiple GFM backbones, including ESM1b and ESM2 variants, on cardiomyopathy (CM) and arrhythmia (ARM) datasets. These fine-tuned models serve as the basis for evaluating adversarial robustness.

Recent studies have underscored the need to evaluate not only data privacy but also the adversarial robustness of genomic AI models—particularly those built on powerful pre-trained architectures. Our findings reveal that GFMs, while strong in generalization, remain vulnerable to attacks that exploit both surface-level perturbations and deeper latent representations. To systematically probe these vulnerabilities, we introduce a dual attack framework that combines perturbation-based and semantic-level adversarial strategies. First, as illustrated in Figure 1c, we use Fast Gradient Sign Method (FGSM) to probe GFM's susceptibility to adversarial sequences by injecting gradient-based noise into variant embeddings, assessing the impact of near-identical sequences on predictions. In addition to FGSM, a **soft prompt attack** is introduced, which operates in the model's embedding space rather than directly modifying input tokens. We consider two variants of the soft prompt attack: a confidence hijack, which reduces the margin between wild-type and variant predictions, and a targeted attack, which specifically pushes benign variants toward pathogenic predictions. As shown in Figure 1d, this method optimizes continuous prompt embeddings that, when prepended to the wild-type and variant inputs, guide the model toward incorrect predictions. Unlike token-level attacks, soft prompts leave the input sequence unchanged but manipulate internal representations to influence decision boundaries. This dual approach enables a comprehensive evaluation of GFM's adversarial robustness by combining input-space perturbations and embedding-space manipulations, revealing the model's susceptibility to both types of attacks.

Across both CM and ARM datasets, all evaluated GFMs exhibited susceptibility to adversarial perturbations, with consistent drops in AUC and AUPR under both input-space (FGSM) and embedding-space (soft prompt) attacks. The soft prompt attack with targeted optimization resulted in the most severe degradation, especially on smaller ESM2 models [4], where AUC and AUPR decreased by up to 10 and 6 percentage points, respectively, with even larger declines observed in the shallow MLM-based ProteinBERT model [5]. Even high-capacity models such as ESM1b [6] and ESM1v [7] were not robust, showing large performance drops under targeted manipulation. In contrast, the confidence hijack variant of the soft prompt attack led to more subtle but consistent degradation, reflecting the sensitivity of learned embeddings to adversarial manipulation. These findings demonstrate the urgent need to evaluate and improve the robustness of GFMs, particularly in clinical genomics where decision boundaries must remain stable under distribution shifts and intentional manipulation. Moreover, they demonstrate a critical limitation in current defense paradigms, which often assume that adversarial perturbations are easily detectable via prediction confidence or input-level cues. Our results suggest that effective defenses must instead operate in latent space—monitoring shifts in embedding geometry and prediction distributions to detect semantically aligned but deceptive adversarial behavior.

Recent advances have positioned GFMs at the center of precision medicine, with increasing use in diagnostic support tools, variant effect prediction, and genome interpretation platforms [8, 9]. These models are often deployed through open-source APIs (e.g., HuggingFace's facebook/esm2), fine-tuned collaboratively across institutions, or integrated into clinical work-

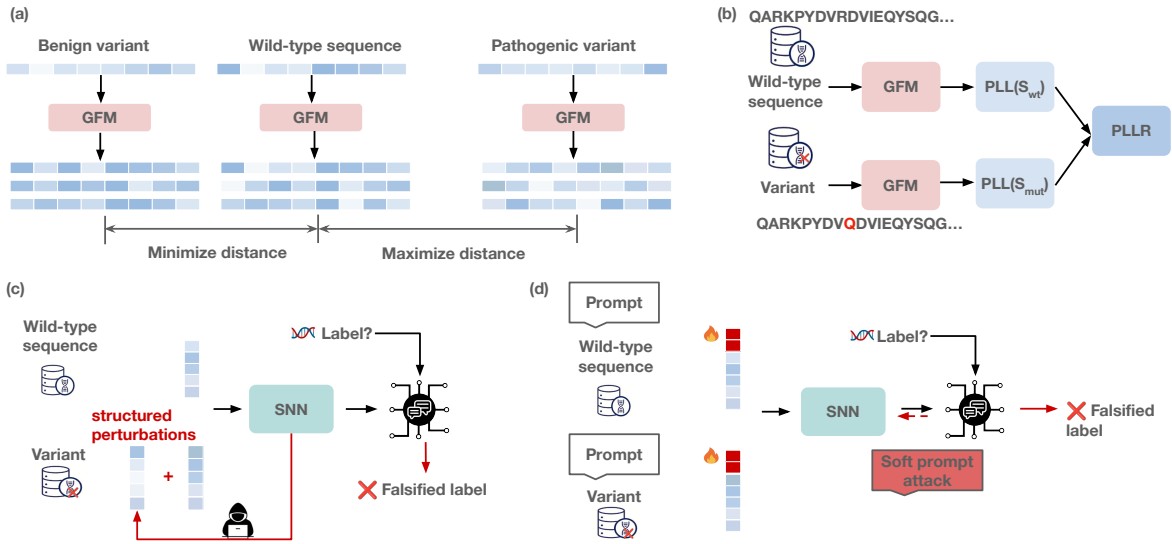

**Figure 1.** Illustration of adversarial sensitivity in GFMs for variant effect prediction. (a) Conceptual schematic showing wild-type and variant embeddings in GFM representation space. Pathogenic variants are expected to maximize distance from the wild-type, while benign variants minimize distance. (b) Overview of PLLR computation using a language model. The PLL for the wild-type and mutant sequence is compared to infer the variant label. (c) FGSM attack perturbs model embeddings to falsify pathogenicity predictions, shifting benign variants to appear pathogenic and vice versa. (d) Soft prompt attack: structured prompts induce the model to shift decision boundaries, leading to adversarial misclassification even when the original label is correct.

flows via cloud-based model endpoints. In such contexts, realistic threat models emerge. An attacker may wish to induce false-positive variant calls to manipulate diagnoses for insurance fraud, research sabotage, or biosecurity evasion [10, 11]. Access could be obtained via model-sharing practices, gradient leakage in federated learning [**zhu2019deep**], prompt injection in downstream pipelines, or through the manipulation of embedding-based model interfaces that accept soft prompts [12]. Because many GFMs operate on pre-tokenized or embedded inputs and often lack strong input sanitization (e.g., they accept variable-length amino acid sequences without constraints), these attacks can bypass traditional input-level verification. Our work mirrors similar threat models in NLP and protein design, where adversarial prompts have been shown to induce targeted model behavior across tasks and modalities [13, 14]. Accordingly, we demonstrate how both gradient-based and prompt-based adversarial strategies can compromise GFMs, even when perturbations are minimal and biologically plausible.

# RESULTS

## Robustness Analysis on the Cardiomyopathy Dataset

To evaluate the impact of adversarial perturbations on variant effect prediction, we performed a robustness analysis using the CM dataset under both clean and FGSM conditions. Figure 2 provides a comprehensive evaluation of the adversarial robustness of PLLR-based variant effect

prediction on the CM dataset.

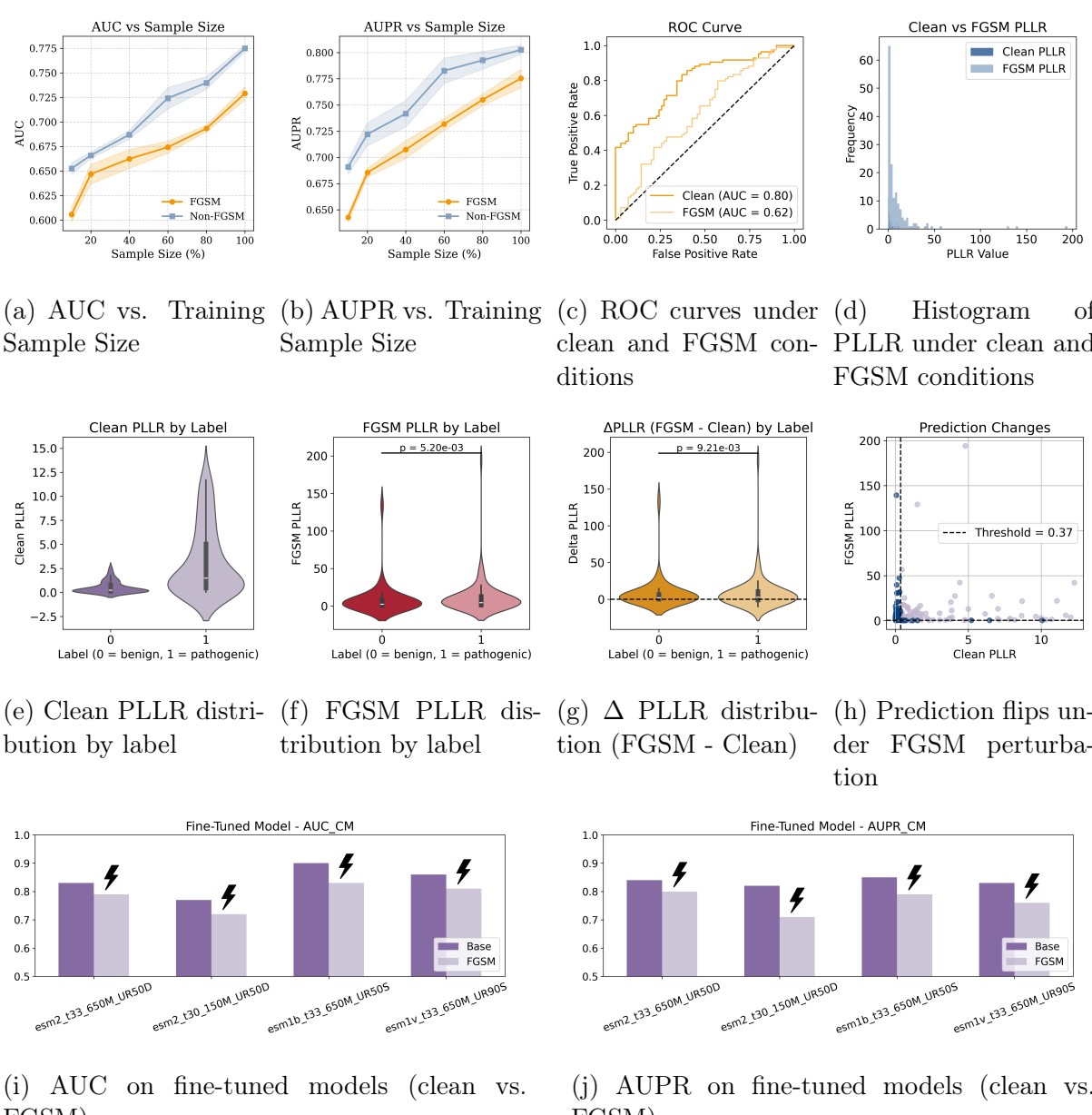

(a) AUC vs. Training Sample Size

(b) AUPR vs. Training Sample Size

(c) ROC curves under clean and FGSM conditions

(d) Histogram of PLLR under clean and FGSM conditions

(e) Clean PLLR distribution by label

(f) FGSM PLLR distribution by label

(g) Δ PLLR distribution (FGSM - Clean)

(h) Prediction flips under FGSM perturbation

(i) AUC on fine-tuned models (clean vs. FGSM)

(j) AUPR on fine-tuned models (clean vs. FGSM)

**Figure 2.** Comprehensive robustness analysis of PLLR-based variant effect prediction on the cardiomyopathy (CM) dataset. (a–b) Performance degradation under FGSM across varying sample sizes. (c–d) ROC and histogram views show confidence collapse. (e–g) Violin plots reveal label-wise distribution shifts in PLLR. (h) Threshold crossing plot shows adversarially flipped predictions. (i–j) AUC and AUPR performance degradation across fine-tuned model variants under FGSM attack. Lightning bolt icons indicate FGSM-perturbed evaluations.

Figure 2 (a) shows that AUC scores consistently improve as the training set size increases. However, across all sampling percentages, the model evaluated with FGSM-perturbed inputs exhibits a clear performance drop relative to its clean counterpart. For instance, at full data (100%), the clean model reaches an AUC of 0.78, while the FGSM model achieves only 0.73. This gap persists even at lower data fractions, revealing that adversarial inputs cause systematic degradation in discriminative power. Figure 2 (b) displays a similar pattern in AUPR, reflecting the precision-recall tradeoff. Without FGSM, the model reaches a peak AUPR of over 0.80,

whereas the FGSM-exposed model underperforms by approximately 3–5 percentage points at every sample size. These results reinforce that the PLLR-based scoring is not only sensitive to adversarial noise but also exhibits increasing robustness with larger training sets. In conclusion, the consistent drop in both AUC and AUPR under adversarial evaluation suggests that models trained for variant effect prediction in clinical genomics must be stress-tested with adversarial scenarios to ensure reliability in real-world settings.

Figure 2(c–h) provides a comprehensive view of how FGSM perturbations affect model confidence and prediction stability. The ROC curves in Figure 2(c) show that performance degrades consistently under adversarial perturbation: while the clean evaluation yields an AUC of 0.80, FGSM reduces this to 0.62. This drop reflects a loss in discriminative capacity caused by small, structured input perturbations.

To further understand how prediction scores shift, Figure 2(d) compares the distributions of PLLR values under clean and adversarial conditions. Clean PLLR scores exhibit clear separation, with pathogenic variants having higher values. However, FGSM shifts the entire PLLR distribution downward, compressing the score range and blurring the margin between classes. Figures 2(e) and (f) break down PLLR distributions by label. Under clean conditions (e), pathogenic variants exhibit high PLLR with wide spread, while benign variants are tightly centered at low values. After FGSM, the pathogenic distribution becomes narrower and overlaps more with the benign class in Figure 2(f). This reduction in separation shows that FGSM weakens the signal used to distinguish variant pathogenicity. The degree of PLLR change is quantified in Figure 2(g), which plots the Δ PLLR (FGSM - clean) for each label. Pathogenic variants show a larger and more variable drop in PLLR than benign ones. This asymmetric sensitivity suggests that FGSM has a stronger effect on confident predictions—likely due to steeper gradient directions near high-PLLR regions.

Figure 2(h) visualizes which samples switch predicted labels due to FGSM. Clean vs. FGSM PLLR values are plotted, and purple points mark samples that cross the decision threshold. Many lie just above the threshold under clean conditions and are pushed below it after attack. This illustrates that the model's decision boundary is not robust, especially for borderline cases.

Lastly, to evaluate whether the observed robustness patterns generalize across models, we further analyze AUC and AUPR performance of four fine-tuned variant effect predictors under clean and FGSM conditions. As shown in Figure 2(i–j), FGSM consistently reduces model performance across all architectures, confirming that adversarial vulnerability is not model-specific. The model identifiers shown in Figure 2 (e.g., esm2_t33_650M_UR50D, esm2_t30_150M_UR50D, esm1b_t33_650M_UR50S, and esm1v_t33_650M_UR90S) refer respectively to the ESM2-650M, ESM2-150M, ESM1b-650M, and ESM1v-650M genomic foundation model variants. The performance degradation is most pronounced in the AUPR metric, where even high-capacity models like ESM1b and ESM1v suffer drops of up to 5 percentage points. These results extend our earlier findings by demonstrating that adversarial perturbations universally reduce discriminative power—even in robust fine-tuned models—emphasizing the need for integrated robustness during model development.

Together, these analyses reveal that PLLR-based scoring is vulnerable to adversarial perturbations, particularly for confident pathogenic predictions. The consistent degradation in both AUC and score separability underscores the need for adversarial robustness in genomic language models.

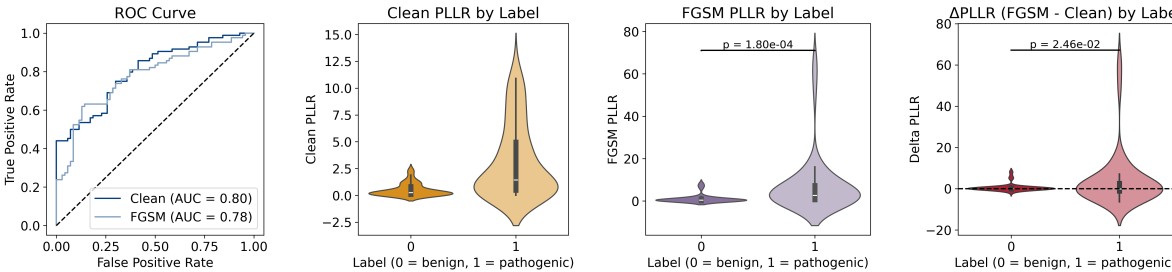

(a) ROC curve: clean vs FGSM conditions

(b) Clean PLLR distribution by label

(c) FGSM PLLR distribution by label

(d) Δ PLLR (FGSM - Clean) by label

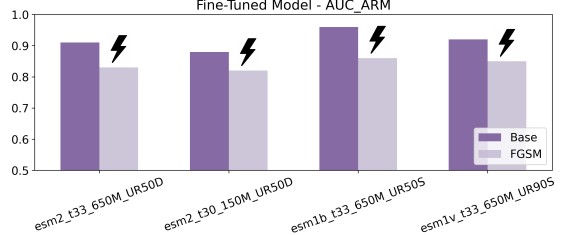

(e) AUC on fine-tuned models (clean vs. FGSM)

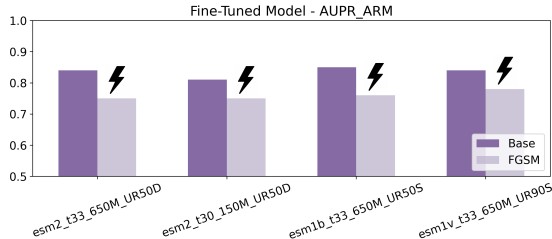

(f) AUPR on fine-tuned models (clean vs. FGSM)

**Figure 3.** Evaluation on the arrhythmia (ARM) dataset under clean and adversarial (FGSM) conditions. (a) ROC performance slightly degrades under FGSM. (b–d) Violin plots show score distribution compression and asymmetric vulnerability across variant labels. (e–f) AUC and AUPR performance degradation across GFMs under FGSM perturbation.

## Robustness Analysis on the Arrhythmia Dataset

We further evaluate adversarial robustness on the ARM dataset, with results shown in Figure 3. The ROC curve in Figure 3(a) indicates a moderate drop in classification performance under FGSM perturbation, where AUC declines from 0.80 to 0.78. Although this drop is smaller than on the CM dataset, it still reflects a sensitivity to input perturbations.

Figures 3(b) and (c) show the PLLR distributions stratified by ground-truth labels. Under clean conditions (b), the model assigns clearly separable scores: benign variants center tightly near zero, while pathogenic variants occupy a broader, higher range. However, FGSM reduces the variance and separation in the pathogenic distribution (c), compressing the score space and pushing many predictions closer to the decision boundary.

The ΔPLLR violin plot in Figure 3(d) quantifies the impact of FGSM perturbations per label. Pathogenic variants (label=1) show a broader and more negative shift in PLLR, confirming that confident high-scoring predictions are more easily degraded. In contrast, benign predictions remain relatively stable. This pattern suggests that the model's internal representations for pathogenic variants are more brittle under adversarial conditions, consistent with findings from the CM dataset. Figures 3(e) and (f) summarize the impact of FGSM perturbations on AUC and AUPR across multiple GFMs evaluated on the ARM dataset. Across all model architectures, FGSM leads to a consistent drop in performance, confirming its effectiveness in disrupting decision boundaries even in models trained for clinical variant interpretation. Smaller models such as ESM2 with 150M parameters experience the largest degradation, while higher-capacity

models like ESM1b and ESM1v, although more resilient, still exhibit large performance loss. These results reinforce that no model is inherently robust and that adversarial perturbations can significantly impair the predictive reliability of GFMs in sensitive clinical contexts such as arrhythmia variant classification.

Overall, these results demonstrate that although the ARM model is slightly more robust than the CM model in AUC terms, it remains susceptible to adversarial degradation, especially for pathogenic predictions with initially high confidence.

## Targeted Soft Prompt Attack (Benign→Pathogenic) on CM

To ensure statistical robustness, we extend our analysis with confidence intervals and bootstrap-based significance testing. Specifically, we report 95% confidence intervals on $\Delta$PLLR in Figures 4(b,f), and use bootstrapping (1,000 samples) to compute empirical confidence intervals on benign $\Delta$PLLR in Figures 4(a,e). Statistical tests (paired $t$-tests and bootstrapped CIs) are applied separately to CM and ARM datasets to validate the significance and asymmetry of the targeted soft prompt attack.

We evaluate the effectiveness of a targeted soft prompt attack designed to selectively elevate the PLLR of benign variants, forcing them to be misclassified as pathogenic. Figure 4(a) shows that benign predictions shift significantly after the attack, with the PLLR distribution moving rightward while pathogenic scores remain largely unchanged. This observation is statistically supported by both a paired $t$-test ($p = 9.23 \times 10^{-4}$) and a bootstrap confidence interval on benign $\Delta$PLLR (mean = 0.156, 95% CI = [0.071, 0.250]), confirming a consistent and meaningful shift in prediction behavior. To quantify per-class vulnerability, Figure 4(b) presents the change in PLLR ($\Delta$PLLR) across labels, with error bars indicating 95% confidence intervals around the mean. The attack exhibits a clear asymmetric effect: benign variants experience a significant positive shift ($\Delta$PLLR = 0.16 ± 0.09), while pathogenic variants show no meaningful change ($\Delta$PLLR = -0.07 ± 0.09). This confirms that the soft prompt optimization objective selectively targets benign variants, leaving the pathogenic class relatively unaffected.

The impact on downstream decision-making is illustrated in Figure 4(c), which plots the precision-recall (PR) curve before and after the attack. The area under the PR curve (AUPR) drops from 0.69 to 0.65, indicating a reduced ability to separate benign from pathogenic variants, driven by an increased false positive rate.

Finally, Figure 4(d) provides a sorted waterfall plot of $\Delta$PLLR for benign variants. While the majority of benign variants exhibit modest changes, a subset shows large increases in PLLR, exceeding +1.0. These outliers demonstrate that the soft prompt is particularly effective at amplifying confidence on select benign cases, resulting in focused prediction flips.

Together, these results reveal the asymmetric nature of the targeted attack: it preserves the pathogenic class while systematically degrading the model's trustworthiness on benign samples. This highlights the importance of adversarial robustness measures in variant effect prediction.

## Targeted Soft Prompt Attack (Benign→Pathogenic) on ARM

To evaluate generalizability, we applied the same targeted soft prompt attack to the ARM dataset. The attack again targets benign variants, aiming to increase their PLLR and induce misclassification. Figure 4(e) shows the PLLR distribution before and after the attack for both benign and pathogenic variants. As in CM, benign variants exhibit a clear rightward shift in

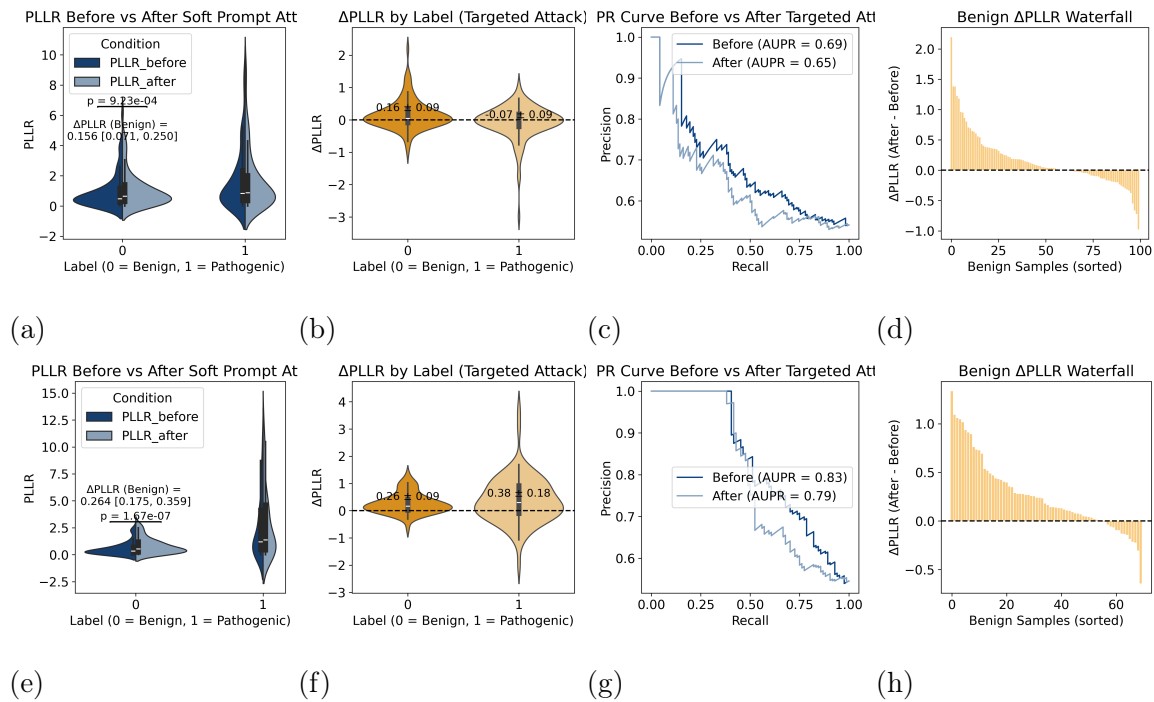

**Figure 4.** Targeted soft prompt attack results. (a–d) CM dataset; (e–h) ARM dataset. (a, e) PLLR before vs. after. (b, f) ΔPLLR by label. (c, g) PR curves. (d, h) Benign ΔPLLR waterfall plots. All results shown are based on the ESM2-650M model.

PLLR, while the pathogenic distribution remains largely unchanged. This is statistically supported by a paired $t$-test on benign variants ($p = 1.67 \times 10^{-7}$) and a bootstrap confidence interval for benign ΔPLLR (mean = 0.264, 95% CI = [0.175, 0.359]), both of which confirm the robustness of the attack effect. The ΔPLLR violin plot in Figure 4(f) further validates this asymmetric behavior, with 95% confidence intervals overlaid on the mean shifts. Benign variants again show a substantial positive change (ΔPLLR = 0.26 ± 0.09), whereas pathogenic variants exhibit a smaller positive but still meaningful shift (ΔPLLR = 0.38 ± 0.18). Compared with the CM dataset, the ARM results suggest that pathogenic variants in ARM are slightly more susceptible to soft prompt perturbations, although the targeted attack still most strongly affects the benign class. Together, these findings demonstrate that the targeted soft prompt attack generalizes across datasets, consistently inducing asymmetric PLLR shifts that disproportionately affect benign predictions.

Figure 4(g) shows that the AUPR drops from 0.83 to 0.79, reflecting degraded performance due to the increased false positive rate. This consistent performance drop across datasets demonstrates that the attack generalizes. Finally, Figure 4(h) presents the ΔPLLR waterfall plot for benign ARM variants. Again, we observe that while many benign samples shift modestly, a subset displays extreme increases in PLLR—evidence that the soft prompt focuses on high-impact samples even in new distributions.

These results demonstrate that the targeted attack is not only effective in CM but also generalizes to ARM, indicating a broader vulnerability in model confidence estimation across datasets.

## Comparative Analysis of Attack Effectiveness

To quantify and compare the impact of different adversarial strategies, we evaluate three attack methods—FGSM, SPA_Confidence Hijack (soft prompt attack with confidence hijack), and SPA_Targeted Attack—across two benchmarks (CM and ARM) and two evaluation metrics (AUC and AUPR). Results are summarized in Figure 5(a–d). For clarity, full model names (e.g., `esm2_t30_150M_UR50D`) are abbreviated in text (e.g., ESM2-150M) following standard naming conventions. On both CM and ARM, we observe that all attack methods reduce model performance to varying degrees. FGSM consistently causes moderate drops in both AUC and AUPR, reflecting its role as a standard untargeted perturbation baseline. The confidence hijack attack (SPA_Confidence Hijack) leads to more pronounced performance degradation, especially in AUPR (Figure 5(b, d)), confirming its effectiveness at collapsing the confidence margin between classes. The most severe performance degradation is observed under the SPA_Targeted Attack condition, which directly optimizes the PLLR of benign samples to mimic pathogenic profiles. This method consistently yields the lowest AUC and AUPR across all model variants, particularly on the ARM benchmark (Figure 5(c–d)). This sharp performance decline highlights the increased vulnerability of models to targeted soft prompt optimization, especially under low-signal or low-data settings.

Overall, the comparative results demonstrate that soft prompt-based attacks can be more effective than input-space perturbations (like FGSM), particularly when targeted or designed to manipulate model confidence.

To examine whether adversarial vulnerabilities are unique to large-scale genomic foundation models or extend to other sequence models, we include ProteinBERT—a masked language modeling (MLM)-based architecture—as a comparative baseline. Unlike ESM models, which are trained on massive protein corpora using transformer-based autoregressive or masked objectives at scale, ProteinBERT represents a smaller, MLM-style model with more constrained capacity and pretraining scope. As shown in Figure 5, ProteinBERT demonstrates the lowest baseline performance across all models and suffers the most conspicuous degradation under adversarial attack, particularly under the SPA_Confidence Hijack condition. The model's susceptibility to embedding-space manipulation suggests that robustness is not solely a function of model scale but also of pretraining strategy and architectural depth. These findings show that MLM-based protein models, while effective in clean settings, are highly brittle under adversarial prompting, reinforcing the broader need for robustness evaluation across the full spectrum of sequence models used in genomics.

## Comparative Analysis Under Different Adversarial Attack Strategies

To further contextualize the robustness of GFMs, we evaluated four additional adversarial attacks: Carlini & Wagner (C&W) [15], DeepFool [16], Projected Gradient Descent (PGD) [17], and Boundary Attack [18], alongside FGSM and soft prompt attacks. For clarity, full model names (e.g., `esm2_t30_150M_UR50D`) are abbreviated in text (e.g., ESM2-150M) following standard naming conventions for Tables 1 and 2. As shown in Tables 1 and 2, the newly introduced gradient-based attacks, PGD, C&W, and DeepFool, produced significantly stronger degradation than FGSM across all model types. On the CM dataset, these attacks reduced AUC by over 30 percentage points on some models. For example, PGD reduced the AUC of ESM2-650M from 0.740 to 0.470, and DeepFool reduced ESM1b from 0.810 to 0.492. Similar trends were observed

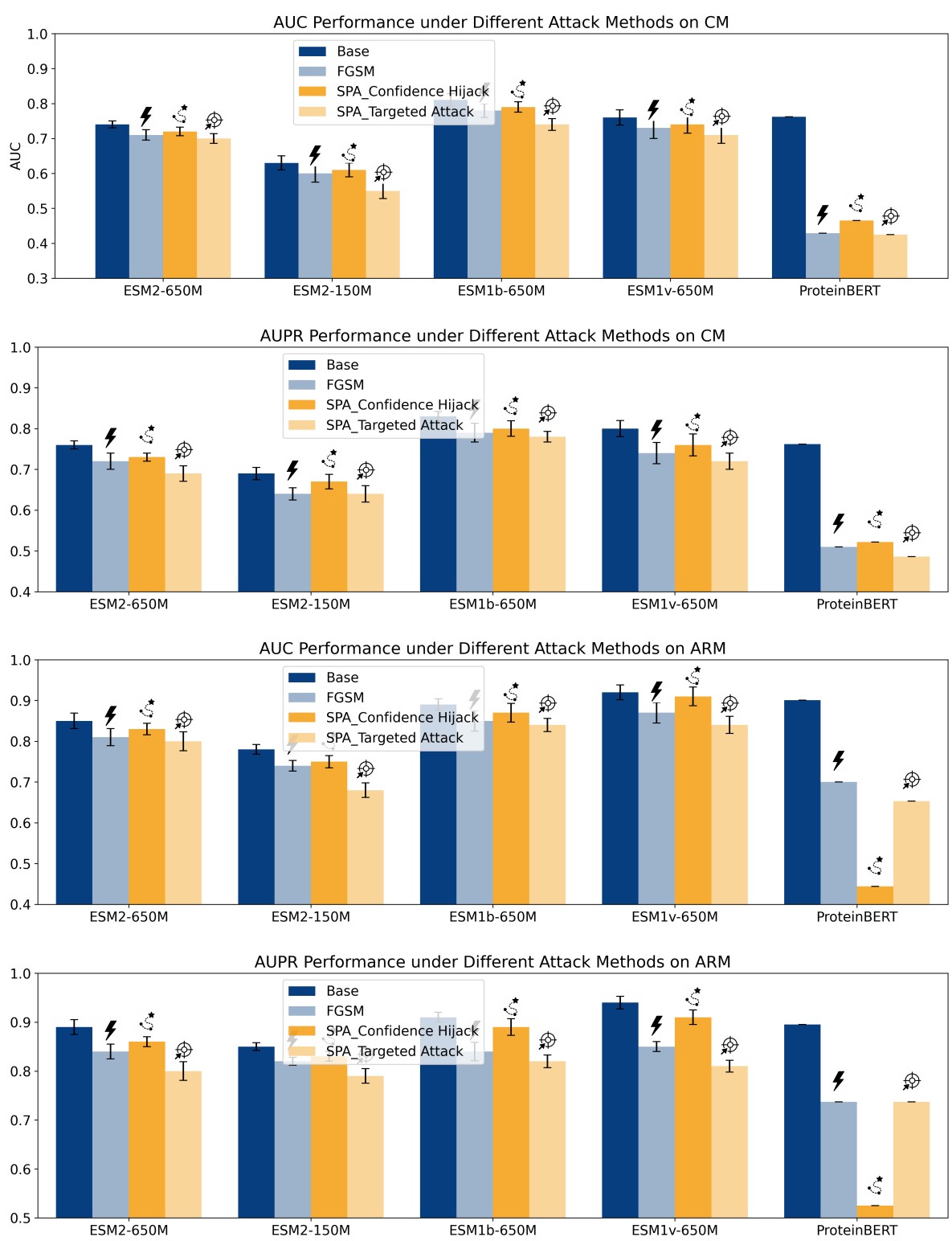

**Figure 5.** Performance under different adversarial attack methods across CM and ARM datasets. (a) AUC on CM. (b) AUPR on CM. (c) AUC on ARM. (d) AUPR on ARM. FGSM, SPA_Confidence Hijack, and SPA_Targeted Attack all degrade model performance, with the targeted attack showing the strongest effect.

**Table 1.** AUC scores on the CM dataset under different attack strategies.

| Model | ESM2-650M | ESM2-150M | ESM1b-650M | ESM1v-650M | ProteinBERT |
|---|---|---|---|---|---|
| Base | 0.740 | 0.630 | 0.810 | 0.760 | 0.762 |
| FGSM | 0.710 | 0.600 | 0.780 | 0.730 | 0.429 |
| DeepFool | 0.490 | 0.585 | 0.492 | 0.471 | 0.424 |
| C&W | 0.494 | 0.593 | 0.497 | 0.479 | 0.430 |
| PGD | 0.470 | 0.588 | 0.482 | 0.465 | 0.426 |
| Boundary Attack | 0.710 | 0.585 | 0.764 | 0.718 | 0.425 |
| SPA_Confidence Hijack | 0.720 | 0.610 | 0.790 | 0.740 | 0.465 |
| SPA_Targeted Attack | 0.700 | 0.550 | 0.740 | 0.710 | 0.425 |

**Table 2.** AUC scores on the ARM dataset under different attack strategies.

| Model | ESM2-650M | ESM2-150M | ESM1b-650M | ESM1v-650M | ProteinBERT |
|---|---|---|---|---|---|
| Base | 0.850 | 0.780 | 0.890 | 0.920 | 0.901 |
| FGSM | 0.810 | 0.740 | 0.850 | 0.870 | 0.700 |
| DeepFool | 0.676 | 0.624 | 0.640 | 0.661 | 0.663 |
| C&W | 0.682 | 0.631 | 0.645 | 0.665 | 0.680 |
| PGD | 0.674 | 0.651 | 0.647 | 0.669 | 0.650 |
| Boundary Attack | 0.805 | 0.702 | 0.844 | 0.859 | 0.642 |
| SPA_Confidence Hijack | 0.830 | 0.750 | 0.870 | 0.910 | 0.444 |
| SPA_Targeted Attack | 0.800 | 0.680 | 0.840 | 0.840 | 0.653 |

on ARM, where PGD reduced ProteinBERT's AUC from 0.901 to 0.650, while DeepFool and C&W dropped it further to 0.663 and 0.680, respectively. These results confirm that PGD and C&W are substantially more effective than FGSM, and that DeepFool—though designed for minimal perturbation—can still induce significant errors in GFM predictions. Interestingly, the black-box decision-based Boundary Attack also achieved comparable degradation on smaller models such as ESM2-150M and ProteinBERT, indicating that even models without gradient access remain vulnerable. Taken together, these findings reveal that both shallow and deep GFMs are susceptible to a range of adversarial attacks, and that attack strength correlates with model capacity and architecture.

## Cross-Model Transferability of Adversarial Soft Prompts

For cross-model transfer, we froze the adversarial prompt learned on a source GFM after full fine-tuning and applied it to other GFMs using their respective adversarially fine-tuned checkpoints, with no further updates. This setting tests whether adversarial perturbations in the embedding space exploit shared vulnerabilities across different model architectures, analogous to universal adversarial triggers in NLP [19].

Table 3 and Table 4 present AUC scores on the CM and ARM benchmarks, respectively. Each row corresponds to a target model being evaluated, while each column indicates the source model from which the prompt was trained. A random prompt baseline is also provided. We observe consistent AUC degradation when using soft prompts transferred across models. For example, a prompt trained on ESM2-150M reduces the AUC of ESM1b-650M from 0.809 (random) to 0.741 on CM, and from 0.899 to 0.832 on ARM. These results suggest that adversarial soft prompts possess partial transferability across GFM architectures, demonstrating the results of shared inductive biases and latent vulnerabilities.

**Table 3.** Cross-model soft prompt transfer on the CM dataset. Rows denote target models, and columns indicate the source models from which the adversarial soft prompts were derived.

| Target Model | ESM2-650M | ESM2-150M | ESM1b-650M | ESM1v-650M | Random |
|---|---|---|---|---|---|
| ESM2-650M | 0.70 | 0.708 | 0.71 | 0.706 | 0.75 |
| ESM2-150M | 0.54 | 0.55 | 0.565 | 0.552 | 0.635 |
| ESM1b-650M | 0.738 | 0.741 | 0.74 | 0.743 | 0.809 |
| ESM1v-650M | 0.696 | 0.716 | 0.721 | 0.71 | 0.763 |

**Table 4.** Cross-model soft prompt transfer on the ARM dataset. Rows denote target models, and columns indicate the source models from which the adversarial soft prompts were derived.

| Target Model | ESM2-650M | ESM2-150M | ESM1b-650M | ESM1v-650M | Random |
|---|---|---|---|---|---|
| ESM2-650M | 0.800 | 0.811 | 0.795 | 0.784 | 0.854 |
| ESM2-150M | 0.678 | 0.680 | 0.688 | 0.692 | 0.782 |
| ESM1b-650M | 0.841 | 0.832 | 0.840 | 0.821 | 0.899 |
| ESM1v-650M | 0.844 | 0.822 | 0.835 | 0.840 | 0.921 |

## Gene-Level Vulnerability Analysis Under PGD Attacks

To investigate gene-specific adversarial vulnerabilities, we applied a Gaussian Mixture Model (GMM) to cluster perturbed PLLR values following PGD attacks. We then quantified the mismatch between the GMM-assigned cluster and the true pathogenicity label for each variant, using this mismatch rate as a proxy for prediction instability under adversarial perturbation. As shown in Figure 6, we conducted this analysis under two PGD configurations: 5 steps and 10 steps. Across both settings, the gene *MYH7* consistently exhibited the highest number of mismatches (63 under PGD-5 and 64 under PGD-10), indicating strong susceptibility to adversarial perturbations and substantial model fragility in its interpretation. *MYBPC3* was the second-most affected gene in both analyses, reinforcing that variants in commonly mutated cardiomyopathy genes are especially sensitive to adversarial manipulation. Several genes showed step-dependent vulnerability patterns. For example, *DES*, *PLN*, and *LAMP2* appeared among the most mismatched genes under PGD-5 but were less prominent under PGD-10, suggesting that some adversarial errors resolve or shift with extended optimization. Conversely, *MYL3* only emerged as highly vulnerable under PGD-10, indicating that stronger attacks can uncover subtler weaknesses in variant-level representations. These trends are illustrated in Figures 6 (a) and 6 (b), which show the mismatch counts for the top genes under each setting. To further investigate how adversarial inputs alter embedding geometry, we include a UMAP visualization of variant-level embeddings before and after targeted soft prompt attacks in Figure 6(c). This panel projects clean and confidence hijacked representations, color-coded by ground truth label, and shows benign → pathogenic label flips. Variants from genes such as *MYL3*, *SCN5A*, and *TNNT2* visibly shift toward the pathogenic region in embedding space, consistent with adversarial goal alignment. These transitions provide geometric evidence that the soft prompt attacks move benign representations across decision boundaries, reinforcing the plausibility of the observed classification flips in panels (a–b). Overall, these results demonstrate the importance of examining adversarial robustness not only in aggregate but also at a gene-specific level. Genes that are disproportionately destabilized by adversarial inputs may benefit from improved calibration, targeted data augmentation, or the incorporation of biological priors to enhance the reliability of genomic foundation models deployed in clinical genomics.

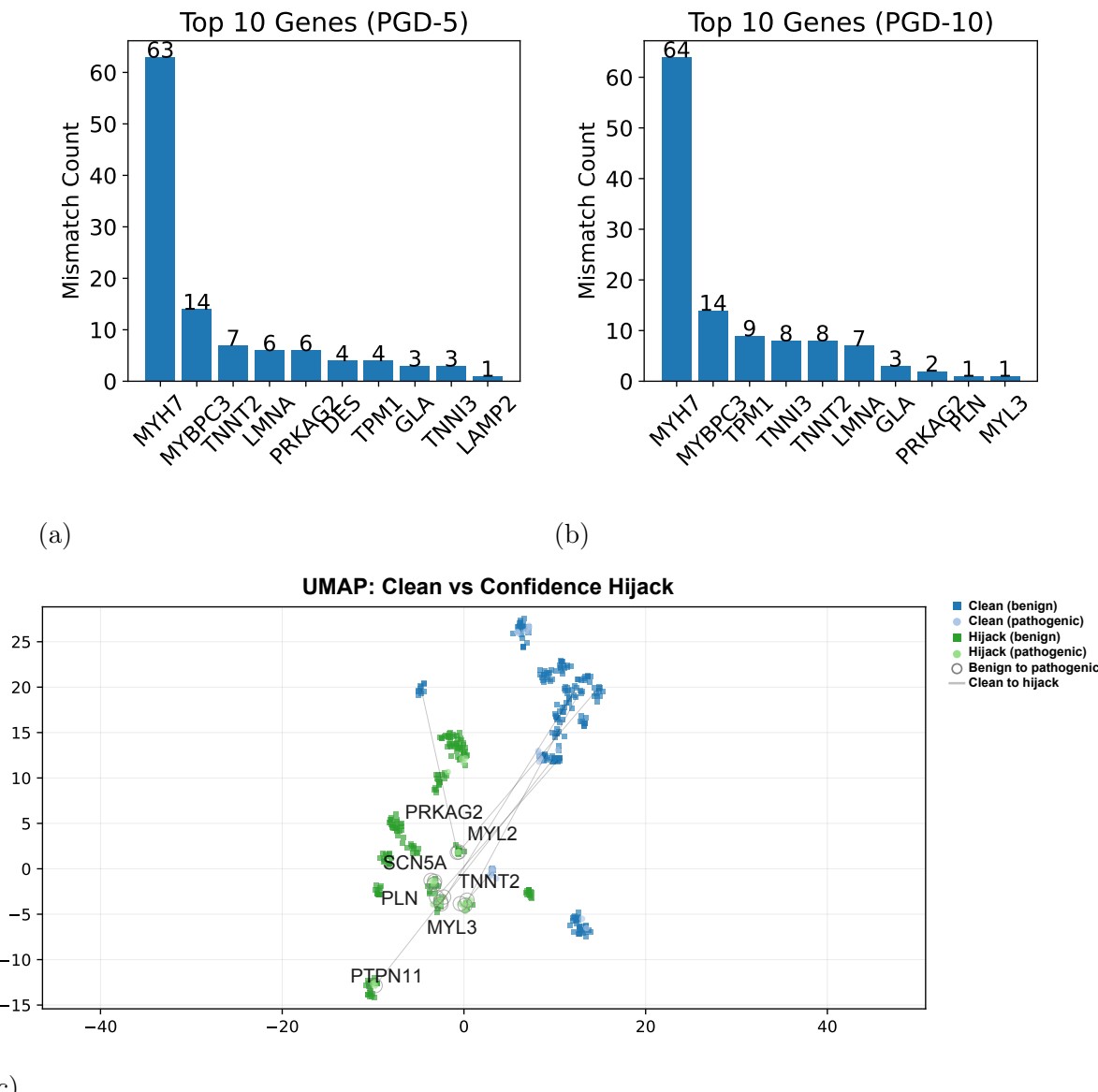

(a)

(b)

(c)

**Figure 6. Gene-level mismatch counts under PGD attacks.** (**a**) Mismatch count per gene between true labels and GMM-clustered predictions under a 5-step PGD attack. (**b**) Same analysis performed under a 10-step PGD attack. Genes with high mismatch counts reflect greater adversarial instability in model predictions for variants affecting those genes. (**c**) UMAP projection of clean vs. soft-prompt (confidence hijack) embeddings, with benign → pathogenic flips highlighted. Labeled genes (e.g., *TNNT2*, *MYL3*, *SCN5A*) show clear shifts toward the pathogenic region under adversarial prompting, suggesting geometry-aligned decision boundary vulnerabilities.

## Clinical Robustness Metrics Under PGD Attacks

To evaluate model reliability in clinically relevant regimes, we computed calibration- and decision-focused robustness metrics under both 5-step and 10-step PGD attacks on the CM dataset, including the Brier score, Expected Calibration Error (ECE), and the false positive rate at a fixed true positive rate of 95% (FPR@TPR=0.95). As shown in Table 5, both PGD-5 and PGD-10 substantially degrade calibration across all GFM variants. Under PGD-10, Brier scores range from 0.46 to 0.50 and ECE values from 0.28 to 0.33, indicating severe miscalibration and unreliable confidence estimates. PGD-5 produces similarly elevated Brier scores (0.44–0.51) and ECE values (0.29–0.33), demonstrating that even moderate adversarial perturbations meaningfully disrupt model probability estimates.

Most critically, the decision-level metric FPR@TPR=0.95 reveals a complete collapse of clinical specificity under both attack strengths. For PGD-10, achieving 95% sensitivity results in a false positive rate of 0.99–1.00 across all models, meaning that nearly all benign variants are misclassified as pathogenic. PGD-5 yields an identical outcome, with FPR values of 1.00 for every model evaluated. These results demonstrate that PGD attacks not only undermine discrimination but also severely compromise clinical decision utility in high-sensitivity operational regimes.

**Table 5.** Calibration and decision reliability metrics under PGD-5 and PGD-10 on the CM dataset.

| Attack | Metric | ESM2-650M | ESM2-150M | ESM1b-650M | ESM1v-650M |
|---|---|---|---|---|---|
| PGD-10 | Brier Score | 0.4606 | 0.5040 | 0.4701 | 0.4593 |
| | ECE | 0.2840 | 0.3250 | 0.2984 | 0.2765 |
| | FPR@TPR=0.95 | 1.0000 | 0.9900 | 1.0000 | 1.0000 |
| PGD-5 | Brier Score | 0.4587 | 0.5110 | 0.4831 | 0.4424 |
| | ECE | 0.3210 | 0.3280 | 0.3082 | 0.2884 |
| | FPR@TPR=0.95 | 1.0000 | 1.0000 | 1.0000 | 1.0000 |

## Ablation Studies on FGSM Perturbation Strength and Soft Prompt Length

To assess the sensitivity of our adversarial framework to design choices, we conducted ablations on the FGSM perturbation magnitude $\epsilon$ and the number of soft prompt tokens. Table 6 shows the AUC of ESM2-650M on the CM dataset across a range of FGSM perturbation strengths. Very small perturbations ($\epsilon \leq 0.005$) yield minimal degradation, whereas $\epsilon = 0.01$ produces the strongest and most stable reduction in AUC without introducing optimization instability. This supports our choice of $\epsilon = 0.01$ in the main experiments.

**Table 6.** AUC on the CM dataset under FGSM with different perturbation magnitudes.

| $\epsilon$ | AUC (CM) |
|---|---|
| 0.001 | 0.724 |
| 0.005 | 0.723 |
| 0.010 | 0.710 |
| 0.020 | 0.716 |
| 0.050 | 0.725 |

Table 7 reports the effect of varying the number of soft prompt tokens. Increasing prompt

length strengthens the adversarial attack, decreasing AUC from 0.734 with 5 tokens to 0.582 with 100 tokens. We therefore select 10 tokens in the main experiments as a balanced configuration: long enough to induce meaningful adversarial perturbation, yet short enough to maintain computational efficiency and avoid overfitting the prompt to a specific model.

**Table 7.** AUC on the CM dataset under soft prompt attacks with different prompt lengths.

| Soft Prompt Tokens | AUC (CM) |
| --- | --- |
| 5 | 0.734 |
| 10 | 0.702 |
| 50 | 0.641 |
| 100 | 0.582 |

## DISCUSSION

Despite growing concerns around privacy and robustness in genomic machine learning, most prior efforts have focused narrowly on data anonymity or privacy-preserving mechanisms. For instance, Kuo et al. [20] emphasized the field's primary concern as the protection of genomic identity, particularly in light of widespread services like 23andMe. Techniques such as differential privacy have been shown vulnerable to de-anonymization attacks [21], while others have explored secure data access methods to prevent malicious retrieval [22]. However, these studies primarily target data privacy, not model integrity.

A separate line of work has emerged to explore attacks on the models themselves, with Montserrat and Ioannidis [23] demonstrating gradient-based white-box attacks on deep learning models used for genomic classification. Feature Importance Model-agnostic Black-box Attack (FIMBA) builds on this direction by introducing a black-box, interpretability-driven framework to compromise gene expression models [24]. By leveraging SHAP-based feature importance [25], FIMBA successfully perturbs top-ranked features to degrade classifier accuracy, even without access to model gradients—marking an important advance in model-agnostic adversarial genomics.

Yet, these efforts still operate on conventional architectures (e.g., MLPs [26, 27], CNNs [28]) and predominantly tabular gene expression data. In contrast, our work moves adversarial robustness research into the era of GFMs—large-scale pre-trained transformers capable of generalizing across diverse biological tasks. We propose two types of adversarial attacks:

- **FGSM perturbation:** a token-space attack that introduces gradient-based noise to the input embeddings, simulating small but adversarial changes to the biological sequence.

- **Soft prompt attack:** an embedding-space strategy that prepends learnable prompt tokens to the input, steering the model's predictions without modifying the original sequence.

Unlike FIMBA, which relies on perturbing high-importance input features derived from model interpretability tools, our methods operate at a deeper semantic level, targeting the latent representation space of GFMs. By crafting semantically aligned adversarial inputs—either through optimized gradient perturbations or soft prompt embeddings—we reveal vulnerabilities that are invisible to surface-level feature-based attacks. These latent space manipulations exploit the internal structure of GFMs, where meaning is encoded across multiple dimensions, making the attacks not only more effective but also more informative for understanding model

behavior and failure modes. This aligns with recent findings in computational pathology, where vision transformers [29] were shown to exhibit more robust latent representations than CNNs under adversarial stress, suggesting that semantic structure in embedding space is a key factor in model resilience [30]. However, while that study focused on visual embeddings, our work offers a parallel in the genomic domain by directly targeting and probing vulnerabilities in the latent space of sequence-based foundation models. Additionally, our soft prompt attack mechanism enables task-specific manipulation without retraining or modifying model weights. This novel approach creates a flexible framework to test and stress the robustness of transformer-based models under adversarial prompts—advancing both our understanding of model generalization and our capacity to build defense mechanisms.

FGSM and the soft prompt attack serve complementary roles in this study. FGSM evaluates the model's sensitivity to small, direct perturbations in the input sequence, offering insights into its robustness against minimally altered genomic variants. In contrast, the soft prompt attack targets the model's internal representation space by optimizing a set of trainable prompt embeddings, effectively misleading the model toward confident yet incorrect predictions without modifying the input sequence. This targeted degradation of the confidence margin not only reveals latent vulnerabilities in GFMs, but also anticipates potential failure modes that are difficult to detect through surface-level perturbations. In particular, our observations align with recent work in vision-language models [31], where aligning the confidence distributions between clean and adversarial samples has been proposed as an effective defense strategy. However, unlike prior work that assumes such distributions are already measurable or optimizable [31, 32], our attacks directly create such pathological shifts in confidence from within the input space—offering a valuable adversarial probe that can expose and quantify semantic instability in GFMs. As such, our work not only contributes a new attack mechanism but also lays the groundwork for developing future defenses that operate in latent and distributional space, rather than relying solely on input-level regularization.

In summary, while prior studies have primarily focused on protecting data privacy or exposing vulnerabilities in shallow genomic classifiers, our work systematically advances adversarial robustness research into the realm of GFMs. By introducing both gradient-based and prompt-based attack paradigms, we uncover failure modes at multiple semantic levels—ranging from input perturbations to latent representation manipulations. These insights directly address the emerging security challenges posed by powerful, general-purpose sequence models in genomics, and also provide actionable tools for evaluating the resilience of such models in clinical and biomedical applications.

Adversarial robustness has gained increasing attention in scientific machine learning domains beyond genomics. In drug discovery, models for molecular property prediction and generation have been shown to be vulnerable to perturbations of molecular graphs or SMILES strings, leading to invalid molecules or manipulated outputs [33]. In protein design, recent work has demonstrated that small adversarial changes to amino acid sequences can disrupt predictions of fitness or structure [34]. Prompt-based attacks have also emerged as an effective strategy in LLMs, where soft prompts—trainable embeddings prepended to input sequences—can be adversarially optimized to elicit targeted outputs or reduce model reliability [35]. Our work draws inspiration from this line of research by adapting soft prompt attacks to Genomic Foundation Models, demonstrating that similar vulnerabilities extend to DNA-level sequence models.

Table 8 summarizes key findings from our evaluation of adversarial robustness in GFMs

**Table 8.** Summary of key findings from adversarial robustness evaluation of genomic foundation models across CM and ARM datasets.

| Evaluation Aspect | Observation and Implication |
| --- | --- |
| Adversarial susceptibility (overall) | All GFMs exhibited consistent degradation in AUC and AUPR under adversarial perturbations, indicating a general lack of robustness across both CM and ARM tasks. |
| Impact of FGSM (input-space) | FGSM introduced minor but reproducible performance drops, showing that even simple gradient-based input perturbations can undermine prediction stability. |
| Impact of soft prompt attack (confidence hijack) | This embedding-space attack subtly collapsed confidence margins between wild-type and variant sequences, leading to systematic but less dramatic reductions in classification performance. |
| Impact of soft prompt attack (targeted) | Targeted prompt optimization produced the largest performance degradation, particularly in smaller models, with AUC reductions of up to 10 percentage points. |
| Relative robustness of larger models | Larger architectures such as ESM1b and ESM1v demonstrated better baseline performance and marginally greater resilience, though they remained vulnerable to targeted prompt-based attacks. |
| Relevance to clinical genomics | The observed vulnerabilities demonstrate the necessity of adversarial robustness evaluation in clinical genomic models, where decision reliability is critical for pathogenicity interpretation. |

across CM and ARM datasets. The results highlight model vulnerabilities under different attack strategies and their implications for clinical deployment.

# METHODS

GFMs, including protein language models such as ESM1b, are typically trained using the Masked Language Modeling (MLM) objective. Within this MLM paradigm, specific amino acid residues in protein sequences are "masked" or hidden, and the model is trained to predict the identity of these masked residues. As the model makes these predictions, it produces raw scores or predictions for each potential amino acid that could replace the masked residue, commonly referred to as "MLM logits". The logits predicted by a protein language model for observing the input amino acid $s_i$ at position $i$ given the sequence $s$ are shown in red frames. When passed through an activation function like softmax, these logits provide probabilities over the possible amino acids, guiding the prediction process.

**Pseudo-log-likelihood ratio computation** In order to fine-tune GFMs on each pair of wild-type and mutant sequences, i,e., $s^{\text{WT}}$ and $s^{\text{mut}}$, we create a siamese network with two weight-sharing protein language model branches (shown in Figure 1 (a) and (b)) to update the weights such that the produced MLM logits are both semantically meaningful and can be compared via pseudo-log-likelihood ratio (PLLR). For a sequence $s = s_1, ..., s_L$, the pseudo-log-likelihood is calculated as $\text{PLL}(s) = \sum_{i=1}^{L} \log P(x_i = s_i | s)$, where $L$ denotes the sequence

length, $s_i$ represents the amino acid at position $i$, and $\log P(x_i = s_i | s)$ denotes the log-likelihood predicted by the protein language model when observing amino acid $s_i$ at position $i$ within sequence $s$. The PLLR between the $s^{\text{WT}}$ and $s^{\text{mut}}$ is then computed as:

$$\lambda = |\text{PLL}(s^{\text{WT}}) - \text{PLL}(s^{\text{mut}})|, \tag{1}$$

because a wild-type sequence typically tends to have a higher log-likelihood in a protein language model and a mutation disrupts the protein with a lower log-likelihood.

**Classification objective function**  To perform the classification of pathogenic vs benign genetic variant via the siamese network, we will utilize a binary cross entropy loss. Binary cross entropy, often referred to as logarithmic loss or log loss, penalizes the model for incorrect labeling of data classes by monitoring deviations in probability during label classification. In order to fine-tune the siamese network using binary cross entropy loss, we calibrate the PLLR to a probability $\hat{\sigma}$ between 0 to 1 by: $\hat{\sigma}(\lambda) = 2\sigma(\lambda) - 1$, due to the sigmoid function $\sigma$ is between 0.5 to 1 for $|\text{PLLR}|$ between 0 to $+\infty$. We then fine-tune the siamese network with the binary cross entropy loss as follows:

$$\mathcal{L}_{\text{BCE}} = y \cdot \log(\hat{\sigma}(\lambda)) + (1 - y) \cdot \log(1 - \hat{\sigma}(\lambda)). \tag{2}$$

Thus, the objective is to maximize the PLLR to distinct the MLM logits between $s^{\text{WT}}$ and $s^{\text{mut}}$ if the mutation is pathogenic and vice versa.

## Attack Models

To evaluate the adversarial vulnerabilities of GFMs, we implement two distinct attack strategies: the FGSM and a soft prompt attack. While FGSM introduces direct input-space perturbations, the soft prompt attack operates in the embedding space and includes two variants: a confidence hijack that disrupts decision margins, and a targeted version that specifically shifts benign variants toward pathogenic predictions.

**Fast Gradient Sign Method (FGSM)**

The FGSM [36] is a widely used adversarial attack that generates perturbed inputs by leveraging the gradient of the loss function with respect to the input. Given an original input sequence $s \in \mathcal{A}^L$ and its corresponding label $y \in \{0, 1\}$, FGSM constructs an adversarial input $s_{\text{adv}}$ using the following formula:

$$s_{\text{adv}} = s + \epsilon \cdot \text{sign}(\nabla_s \mathcal{L}(s, y)),$$

where $\mathcal{L}(s, y)$ is the classification loss, $\nabla_s \mathcal{L}$ denotes the gradient of the loss with respect to the input sequence embeddings, and $\epsilon$ is a small scalar controlling the perturbation magnitude. The sign function ensures that the perturbation is aligned with the direction that most increases the loss, while keeping the perturbation norm minimal.

In this study, FGSM is used to evaluate the robustness of the ESM-based variant effect predictor by introducing subtle perturbations to the wild-type and mutant sequences. We apply FGSM to the embedding space of each input and monitor changes in pseudo-log-likelihood ratio (PLLR) and downstream classification outcomes. This allows us to identify cases where

the model is overly sensitive to near-identical input sequences, thereby revealing adversarial weaknesses in genomic prediction tasks.

**Soft Prompt Attack**

To assess the adversarial vulnerability of the GFMs, we implement a soft prompt attack in which a trainable embedding sequence is prepended to both wild-type and mutant sequences. In contrast to prompt-tuning with a frozen model, we allow full model fine-tuning during the attack to maximize adversarial impact. Both the soft prompt and the underlying protein language model are updated during optimization, enabling more effective perturbation of internal representations.

1. **Confidence Hijack Objective.** In the full-class adversarial setting, we aim to mislead the model by increasing its confidence in incorrect predictions across both classes. Specifically, we retain the pseudo-log-likelihood ratio (PLLR) computation $\lambda = |\text{PLL}(s^{\text{WT}}) - \text{PLL}(s^{\text{mut}})|$, and define a calibrated probability $\hat{\sigma}(\lambda) = 2\sigma(\lambda) - 1$, where $\sigma(\cdot)$ is the sigmoid function. For a ground-truth label $y \in \{0, 1\}$, the adversarial objective flips the labels and maximizes the binary cross entropy loss as:

$$\mathcal{L}_{\text{attack}} = (1 - y) \cdot \log(\hat{\sigma}(\lambda)) + y \cdot \log(1 - \hat{\sigma}(\lambda)). \tag{3}$$

This objective explicitly encourages benign variants ($y = 0$) to exhibit high PLLRs (pathogenic-like) and pathogenic variants ($y = 1$) to exhibit low PLLRs (benign-like), thereby increasing misclassification confidence.

2. **Targeted Soft Prompt Attack (Benign→Pathogenic).** In the targeted one-class attack scenario, we optimize the soft prompt solely to misclassify benign variants. Given a mask over benign examples ($y = 0$), we define the attack loss as:

$$\mathcal{L}_{\text{benign}} = -\log(\hat{\sigma}(\lambda)), \quad \text{for } y = 0. \tag{4}$$

This targeted objective forces the model to output high PLLR values for benign variants, thereby mimicking confident pathogenic predictions. No gradients are applied to pathogenic examples, preserving their outputs while selectively increasing false positive rates.

**Optimization and Evaluation** During attack optimization, we jointly update both the soft prompt embeddings and all GFM backbone parameters via gradient descent, allowing full model fine-tuning under the adversarial objective. The model is evaluated under the original labels using ROC AUC, AUPR, and threshold-based metrics to assess the degradation in classification performance caused by the adversarial soft prompt. This setup allows us to isolate the effect of soft input perturbations on model robustness without altering biological content.

## Settings

To evaluate the robustness of our variant effect prediction framework, we implemented two distinct adversarial strategies: FGSM and soft prompt attacks. For the FGSM setup, we applied

perturbations directly to the input embeddings of the GFM using the Fast Gradient Sign Method, with an attack strength of $\epsilon = 0.01$. The gradient was computed with respect to the classification loss, and the perturbed embeddings were then passed through a frozen GFM encoder followed by a trainable classification head. For PGD, we used 5 attack steps with a step size of 0.002 and maximum perturbation $\epsilon = 0.01$, applying projection onto the $\ell_\infty$ ball after each step. For the C&W attack, we optimized the adversarial objective using 1,000 binary search steps and 9 optimization iterations per step, with a confidence parameter of 0. The soft prompt attack prepended $n = 10$ learnable prompt tokens to each input sequence. These prompt embeddings, initialized via Xavier uniform distribution [37], were optimized jointly with the classification head. For both experiments, we trained the models using the Adam optimizer with a learning rate of $1 \times 10^{-4}$ [38] and batch size of 4 over 10 epochs. All attacks are now performed over five independent runs with different random initializations. Additionally, we apply 1,000-sample bootstrap resampling to quantify the stability of the attack effect across samples in Figure 4. The loss function used was binary cross-entropy, and training was conducted on a single A100 GPU. This design enables a direct comparison between perturbation-based (FGSM) and representation-based (soft prompt) adversarial robustness under a consistent training regime.

## Data and Code Availability

This study focuses on clinically relevant variant sets associated with inherited cardiomyopathies (CM) and arrhythmias (ARM). We draw on a previously curated collection of rare missense variants labeled as pathogenic or benign, organized using a cohort-driven framework for disease-specific analysis, as described by Zhang et al. [39]. The dataset is publicly accessible via https://github.com/ImperialCardioGenetics/CardioBoost_manuscript. To supplement this resource, compiled data are available on Zenodo at https://zenodo.org/records/13397296 [40]. The code is publicly available at: https://anonymous.4open.science/r/SafeGenes-9086/.

The statistics for both datasets are shown in Table 9.

**Table 9.** Datasets

|  | Cardiomyopathies | | | | Arrhythmias | | | |
|  | Pathogenic | Benign | VUS | Total | Pathogenic | Benign | VUS | Total |
|---|---|---|---|---|---|---|---|---|
| Train | 238 | 202 | - | 440 | 168 | 158 | - | 326 |
| Test | 118 | 100 | - | 218 | 84 | 79 | - | 163 |
| Total | 356 | 302 | - | 658 | 252 | 237 | - | 489 |

## Acknowledgements

This work was supported by the National Institutes of Health under award numbers NIH R01 LM010098 & U01 AG066833.

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
