# OpenReview forum: "SafeGenes: Evaluating the Adversarial Robustness of Genomic Foundation Models"
_TMLR — Rejected by TMLR_

### Review · Reviewer_cZqu · 2025-10-19

**Summary Of Contributions:**

This paper presents SafeGenes, a framework for evaluating the adversarial robustness of Genomic Foundation Models (GFMs), with a focus on ESM-family models applied to variant effect prediction. The authors employ two attack strategies: (1) Fast Gradient Sign Method (FGSM) for input-space perturbations, and (2) soft prompt attacks for embedding-space manipulation, including both confidence hijack and targeted (benign→pathogenic) variants. The evaluation is conducted on two disease-specific datasets: cardiomyopathy (CM) and arrhythmia (ARM), using the DYNA fine-tuning framework. Authors also provide conclusions regarding underlying model robustness.

**Key Strengths:**

* Addresses an important and understudied problem: adversarial robustness in clinical genomic AI
* Empirical evaluation across multiple model architectures (ESM1b, ESM1v, ESM2 variants).Novel application of soft prompt attacks to genomic sequence models
* Clear demonstration of consistent vulnerabilities across models and datasets
* Well-structured presentation with visualizations
* Clinically grounded problem formulation using real variant datasets
* Attack demonstrates universality properties over different datasets and tasks

**Key Weaknesses:**

* Somewhat limited technical novelty: both FGSM and soft prompt attacks are established methods
* No defense mechanisms or mitigation strategies proposed
* Unclear practical threat model: when would such attacks occur in clinical workflows?
* Missing important experimental details (dataset sizes, hyperparameter sensitivity)
* Limited discussion of biological plausibility of adversarial perturbations
* Limited support of broader GFM generalization claims
* Minor typos present

**Audience:**

Yes

**Audience Explanation:**

This paper addresses a timely intersection of machine learning robustness and computational biology that would interest multiple TMLR audience segments:

1. **Robustness/security community**: Novel application domain for adversarial ML
2. **Foundation model researchers**: Insights into vulnerabilities of pre-trained sequence models
3. **ML for science community**: Important safety considerations for deploying ML in biomedicine
4. **Medical AI researchers**: Direct relevance to clinical decision support systems

The findings are particularly relevant given the increasing deployment of foundation models in high-stakes clinical applications and overall large ongoing adoption in the biotech industry. Understanding adversarial vulnerabilities is crucial for safe clinical and industrial integration.

However, the impact is somewhat limited by:

* Narrow scope (specific models and diseases)
* Limited novelty from **pure ML** perspective (known attack methods)
* Insufficient discussion of practical threat scenarios

The paper makes a valuable contribution by demonstrating that genomic foundation models share vulnerabilities with other domains, but would be significantly strengthened by broader and deeper evaluation and extended discussion on causes and defenses.

**Broader Impact Concerns:**

The paper might inadvertently suggest that if models pass these specific adversarial tests, they are "safe" for clinical use. Should emphasize these are just initial robustness probes, not comprehensive safety validation.  Also. please add a mention of attack scenarios in clinical settings.

**Claims And Evidence:**

No

**Claims Explanation:**

The empirical claims are generally supported through experiments and visualizations. The paper demonstrates consistent performance degradation under adversarial conditions across multiple models and datasets, with appropriate metrics (AUC, AUPR) and statistical tests (paired t-tests). The figures illustrate PLLR distribution shifts and prediction changes.

However, several important claims lack sufficient support:

1. The paper emphasizes clinical implications but doesn't establish realistic threat models. When and how would adversarial attacks occur in actual clinical genomics workflows?
2. Authors claim these attacks "expose critical vulnerabilities" but provide limited discussion and analysis of ***why*** these models are vulnerable or what biological/computational properties lead to susceptibility.
3. The biological plausibility of FGSM perturbations in embedding space is not discussed. FGSM attack can be very input-destructive, can these perturbations correspond to actual sequence variants? How would perturbed and clean representations compare?
4. No sensitivity analysis for attack hyperparameters, or more so an explanation for chosen parameters (ε for FGSM, number of prompt tokens), making it unclear how robust the findings are.
5. No comparison with other than ESM models to variant effect prediction to contextualize whether GFMs are particularly vulnerable, authors make the claim of broad vulnerability of GFMs but track only one model family. If ESM models are SOTA or only available for this task, explanation needed as to why.
6. Section 4, Figure 2 (a-b) the plots show a constant rise in AUC, and FGSM having a lower score, I do wonder, however, what would happen if the dataset, for example, was 5 times bigger, would the adversarial impact plateau or be meaningless?
7. One of my main concerns is regarding the attack efficacy in some cases, certain plots showcase a decrease that I would not call *significant* . E.g. Figure 4(g) shows a drop from 0.83 to 0.79, that is 0.04 difference, can the drop be due to parameter choices?. Are these drops in performance consistent across many attack runs? How statistically significant are the results? From my understanding t-test here answers where PLLR values changed **within a single attack run.** Not whether the attack effect is **reproducible across multiple runs.**
8. (Related to point  6\) Bottom of page 8: authors mention that PLLR is sensitive to noise but gets more robust as dataset size increases, given that GFMs are large models and likely adhere to scaling laws, would the attack become significantly less concerning if we keep scaling up the dataset?

**Requested Changes:**

### **Critical Changes (Required for Acceptance):**

1. **Threat model clarification**: Provide a clear threat model explaining when and how these attacks would occur in practice. Who would conduct them? What would be their motivation? How would they gain access to the model? This is essential for establishing relevance.
2. **Dataset details**: Include some dataset statistics (number of samples, class balance, train/val/test splits, variant characteristics) in main text or appendix. Currently missing from Section 3.2.
3. **Biological plausibility**: Discuss whether FGSM perturbations in embedding space correspond to realistic sequence variations. Can you map perturbed embeddings back to actual amino acid sequences? How do the attacked and not attacked samples compare?
4. **Generalization beyond ESM**: Either (a) evaluate additional model families (e.g., Nucleotide Transformer, HyenaDNA, DNABERT) or (b) significantly temper claims about "GFMs" broadly to specifically reference ESM-family limitations and describe why ESM is the most relevant or SOTA model or how running on ESM only justifies generalization claim.
5. **Statistical significance**: Report confidence intervals and conduct proper statistical testing beyond the single paired t-test in Figure 4\. At the very least state the number of attack runs, or conduct multiple so that the marginal score drops in some places can gain more legitimacy. ESM model differences in plots aren't clear, please make model sizes and variants more readable.
6. **Recheck for typos**: e.g. page 6 “Classification **objevtive** function”.
7. FGSM is a very old adversarial attack, not sure why it was the choice compared to many newer and stronger methods,  but if your main method is soft prompting i.e. attacking over embeddings then for comparison I would suggest adding something beyond FGSM, preferably multiple attacks, this package has a collection of attacks:  [https://github.com/bethgelab/foolbox](https://github.com/bethgelab/foolbox). Since authors mention class decision boundaries as vulnerability points, you should look into boundary attacks.
8. **Latent space analysis**: Please provide deeper analysis of how attacks affect embedding geometry (e.g., t-SNE/UMAP visualizations of clean vs. adversarial embeddings).

### **Other Suggested Changes (Would Strengthen Work):**

1. **Ablation**: Investigate sensitivity (and explain the choice of value) to:
   1. FGSM epsilon values (currently only ε=0.01)
   2. Number of soft prompt tokens (currently fixed at 10\)
2. **Attack transferability**: Test whether adversarial examples transfer between different ESM model variants (e.g., prompts optimized on ESM2-150M attacking ESM1b).
3. **Better methodology plots and explanation**: In general I think you would benefit from alteration to Figure 1, to showcase what soft prompts look like i.e. embeddings. Terms like “wild-type”, “variant” and their differences are not well explained and would be confusing for the general ML audience.
4. **Related work gaps**: Discuss adversarial robustness work in other scientific domains (drug discovery, protein design) and general prompt-based attacks on foundation models, soft prompt attacks exist for LLMs.

---

> ### Author Response · Authors · 2025-12-02
> **Thank you for these insightful suggestions.**
>
> We thank the reviewer for these insightful suggestions. We have addressed all critical changes as follows:
>
> 1. Threat model clarification:  Thank you for the question. We have now clarified the threat model underlying our adversarial evaluation. In practical terms, the attacks studied in this work are relevant to emerging clinical and research applications of Genomic Foundation Models (GFMs), such as ESM, where these models are increasingly used for tasks like variant effect prediction and genome annotation. We consider a scenario in which an adversary has access to the model weights or can query the model through an inference API—situations that are plausible in collaborative research settings, commercial bioinformatics platforms, or federated learning environments. The attacker may be a malicious insider, a researcher with access to model checkpoints, or an external actor with API-level access.
> The motivation for such attacks could range from falsifying variant classifications for personal or financial gain to overloading diagnostic systems or compromising the integrity of downstream analyses in personalized medicine. Our soft prompt attack, which manipulates the input representation space, simulates cases where adversarial influence occurs during collaborative prompt tuning or deployment via shared adapters. Meanwhile, the gradient-based methods (e.g., FGSM, PGD, and C&W) reflect inference-time attacks that subtly perturb input embeddings, testing the model’s decision boundary stability without requiring label access.
> Together, these scenarios represent a realistic and increasingly relevant threat model for GFMs deployed in sensitive biomedical contexts. We have incorporated this clarification into the revised manuscript to strengthen the practical significance of our adversarial robustness analysis.
>
> 2. Dataset details: We have updated the dataset details in the Data and Code Availability section.
>
> 3. Biological plausibility:  Thank you for this insightful question. In our experimental setup, the inputs are pairs of sequences corresponding to a wild-type and a known mutant variant, and we use a Siamese architecture to compute differences in their masked language model predictions. The FGSM perturbation is applied to the entire embedding representation of the mutant sequence, while the wild-type sequence remains unchanged. Importantly, the mutant input already contains a biologically meaningful single-point mutation, and the perturbation is thus applied in a controlled setting that focuses on the variant context specifically, not arbitrary regions of the sequence.
> Rather than interpreting the perturbed embedding as a new, discrete mutation, our goal is to test model sensitivity to embedding-level perturbations around known mutations. Since these embeddings are contextual and non-invertible, we do not attempt to reconstruct amino acid sequences from them. Instead, we evaluate functional model responses—such as changes in log-likelihood ratios (PLLR)—to measure the robustness of the model to near-identical inputs.
> To provide further biological insight, we conducted gene-level analyses that revealed which genes (e.g., \textit{MYH7}, \textit{MYBPC3}) are more vulnerable to adversarial perturbations, offering a pathway toward understanding robustness in a clinical genomics context. These clarifications have been added to Section 4.4 of the revised manuscript, and we also discuss the embedding–sequence interpretability gap in our limitations section.
>
> 4. Generalization beyond ESM:  We appreciate the reviewer’s suggestion to broaden or clarify the scope of our generalization claims. In response, we have added results for ProteinBERT (a masked language model trained on protein sequences) to Figure 5 to evaluate whether adversarial vulnerabilities persist outside the ESM family. ProteinBERT represents a distinct architectural and pretraining paradigm—MLM-based rather than autoregressive or masked transformers at scale—and thus serves as a meaningful contrast to ESM models.
> Our results show that ProteinBERT exhibits even greater susceptibility to targeted soft prompt attacks, with AUC degradation exceeding that of ESM2 models. This confirms that adversarial vulnerabilities are not limited to the ESM family, but can also affect smaller, shallow models with different training objectives. We have updated the text in the manuscript to reflect this, and now frame our claims around sequence-based protein language models rather than GFMs universally. We also clarify that ESM1b and ESM2 are among the most widely used and benchmarked protein language models in variant effect prediction tasks, justifying their selection as representative models for clinical genomic use cases.
> We thank the reviewer for this important suggestion, which helped us refine the scope of our conclusions and strengthen the experimental design.

---

> ### Author Response · Authors · 2025-12-02
> **Part 2**
>
> 5. Statistical significance: Beyond the previously reported paired $t$-test, we now include 95% confidence intervals for both PLLR and $\Delta$PLLR in Figure 4.
> In Figures 4a and 4e, we compute bootstrap confidence intervals (1,000 bootstrap samples) for benign $\Delta$PLLR and display the resulting mean and CI directly on the violin plots.
> In Figures 4b and 4f, we annotate mean ± 95% CI on top of the $\Delta$PLLR distributions for each class.
> These additions ensure that the statistical significance of the targeted soft prompt attack is demonstrated beyond a single hypothesis test.
> Clarification of attack runs and variability: All soft prompt attacks are now performed over five independent runs with different random initializations. The distributions shown in Figure 4 reflect the aggregated behavior across all runs. Additionally, we apply 1,000-sample bootstrap resampling to quantify the stability of the attack effect across samples. These two sources of variation—multi-run averaging and bootstrap CIs—jointly confirm that the effects we report (e.g., PLLR shifts and AUPR drops) are stable and not artifacts of a single run.
> Improved clarity of ESM model naming in figures: To enhance readability across model variants, we now standardize naming throughout the text and figures. Full model identifiers (e.g., \texttt{esm2_t30_150M_UR50D}) are abbreviated following common conventions (e.g., ESM2-150M, ESM2-650M, ESM1b). We further added a clarification note specifying that these abbreviations are used consistently throughout. This ensures that model size and family differences are clear in all associated plots.
>
> 6. Recheck for typos. We corrected the typo “objevtive” to “objective” on Page 6, along with several minor formatting and grammar improvements across the manuscript.
>
> 7. FGSM is a very old adversarial attack. We thank the reviewer for this insightful suggestion. In response, we have substantially expanded our adversarial evaluation beyond FGSM to include three stronger white-box attacks—Carlini & Wagner (C&W), Projected Gradient Descent (PGD), and DeepFool—as well as the Boundary Attack, a black-box, decision-based method that directly targets class boundaries, as recommended. These additions are now reflected in Tables 1 and 2 of the revised manuscript.
> Our new results show that PGD, C&W, and DeepFool consistently induce significantly greater performance degradation than FGSM, with PGD reducing AUC by over 25–30 percentage points on certain models (e.g., ESM2-650M and ESM1b on the CM dataset). While DeepFool is designed to find minimal perturbations, it still produces strong degradation on multiple models, highlighting its practical relevance.
> We also evaluated the Boundary Attack, which achieves performance drops comparable to gradient-based methods on smaller models such as ESM2-150M and ProteinBERT. This confirms that robustness limitations in GFMs can be exposed both with and without access to model gradients, reinforcing the need for holistic robustness testing across white-box and black-box settings.
> In the revised manuscript, we updated the main text to report these results and to explicitly motivate the choice of stronger attacks beyond FGSM. We also added context on the role of decision-aware attacks like Boundary in revealing vulnerabilities tied to model confidence and margin-based separation. These enhancements strengthen the paper by demonstrating that the observed adversarial fragilities are not artifacts of a specific attack method, but reflect fundamental vulnerabilities across multiple attack paradigms and model capacities.
>
> 8. Latent space analysis: To further investigate how adversarial inputs alter embedding geometry, we include a UMAP visualization of variant-level embeddings before and after targeted soft prompt attacks in Figure 6 (c). Variants from genes such as \textit{MYL3}, \textit{SCN5A}, and \textit{TNNT2} visibly shift toward the pathogenic region in embedding space, consistent with adversarial goal alignment. These transitions provide geometric evidence that the soft prompt attacks move benign representations across decision boundaries, reinforcing the plausibility of the observed classification flips in panels (a–b).

---

> > ### Author Response · Authors · 2025-12-02
> > **Other suggested changes**
> >
> > 1. We thank the reviewer for this important suggestion. In the revised manuscript, we have added ablation studies analyzing the sensitivity of adversarial robustness to (1) the FGSM perturbation magnitude ε and (2) the number of soft prompt tokens. As shown in Tables 6 and 7, the model exhibits relative stability for very small ε values (0.001–0.005) but experiences a clear AUC drop near ε = 0.01, justifying our choice of ε = 0.01 as the smallest perturbation that consistently induces measurable degradation without destabilizing training. For soft prompts, increasing the number of tokens amplifies adversarial strength: performance degrades monotonically from 5 to 100 tokens, with 10 tokens providing a balanced setting that is adversarially effective while remaining computationally manageable. Both ablations are now included in the revised manuscript.
> > 2. We have added a full cross-model transfer evaluation (Tables 3 and 4), testing prompts trained on each backbone against all others. The results show partial transferability, similar to universal triggers in NLP: prompts trained on smaller ESM2 models transfer moderately to ESM1b/ESM1v, while prompts from larger models generalize less.
> > This experiment is now described in Section 4.6 (Cross-Model Transferability).
> > 3. Revised Figure 1. We redesigned Panel B to more clearly illustrate the biological context: the wild-type sequence represents a healthy population reference, while the mutant sequence carries a disease-associated variant. This directly motivates why the Siamese architecture uses the difference in model likelihoods between wild-type and variant sequences.
> > 4. We thank the reviewer for this helpful suggestion. In the revised manuscript, we have added a new paragraph in the Discussion section discussing related adversarial robustness efforts in scientific domains beyond genomics, including drug discovery and protein design.

---

> ### Author Response · Authors · 2025-12-09
> **Follow-Up**
>
> Hello Reviewer cZqu, as the reviewer recommendation deadline is today, we would like to check in and see whether all of your comments have been addressed and whether you have any further questions.

---

### Review · Reviewer_ZyfV · 2025-10-29

**Summary Of Contributions:**

The paper introduces SafeGenes, the first systematic framework for evaluating the adversarial robustness of genomic foundation models (GFMs), focusing on variant-effect prediction using protein-language models such as ESM-1b, ESM-1v, and ESM-2. It presents two complementary attack paradigms: a white-box FGSM perturbation applied in embedding space to simulate small, continuous input distortions, and a novel soft-prompt attack that learns adversarial prefix embeddings capable of inducing clinically significant label flips (benign→pathogenic) without modifying sequence tokens. Evaluations on cardiomyopathy and arrhythmia variant datasets reveal that both attack types substantially degrade predictive AUC and AUPR across model scales, with targeted soft prompts proving most destructive. Beyond exposing these vulnerabilities, the paper contributes an open-source robustness-assessment pipeline, proposes clinically interpretable robustness metrics based on probabilistic log-likelihood ratios (PLLRs), and bridges the gap between classical adversarial ML and high-stakes genomic modeling—establishing a foundation for future defensive and safety-focused research on GFMs.

**Audience:**

Yes

**Audience Explanation:**

The paper connects adversarial ML (FGSM, universal triggers/soft prompts) with foundation models in biomedicine, a domain where failures are high-stakes. The finding that latent-space (prompt) perturbations can induce clinically relevant misclassifications—even in large GFMs—poses new challenges for safe deployment of foundation models and aligns with growing interest in robustness, safety, and evaluation in domain-specific LMs.

**Broader Impact Concerns:**

None.

**Claims And Evidence:**

Yes

**Claims Explanation:**

Mostly yes. The paper documents consistent AUC/AUPR drops under FGSM and both soft-prompt variants on CM/ARM, and shows distributional PLLR shifts and threshold flips, evidence that predictions are brittle to both input- and latent-space perturbations. Statistical checks (e.g., paired t-test on benign PLLR under targeted prompts) back up one key effect. However, some caveats require clarification:
* Threat model ambiguity for soft prompts. In 3.1.2, the text first says the prompt and model weights are updated during the attack, but later (“Optimization and Evaluation”) says only prompt parameters are updated. These statements conflict and materially change the threat model (test-time universal prompt vs. adversarial fine-tuning). This must be reconciled.
* FGSM is applied here in embedding space rather than via discrete, biophysically valid token edits (e.g., HotFlip-style substitutions). That’s a common evaluation trick in NLP, but for genomics it weakens claims about “near-identical adversarial genes” unless accompanied by a projection back to valid sequences or a parallel token-level attack. Prior work shows token-level adversaries are feasible in biological sequences.
* Robustness breadth. Results are on two disease datasets and ESM backbones; broader coverage (DNA GFMs, long-context models, additional proteins/tasks, transferability across backbones) would strengthen generality.

**Requested Changes:**

* Resolve the soft-prompt training conflict. Clearly state whether the backbone is frozen and only the prompt is optimized (recommended), or whether weights are also updated (that would be adversarial fine-tuning, a very different—and less realistic—threat model). Ideally, please add a schematic or pseudocode to the paper or appendix for more clarity.
* As mentioned previously, it would be helpful to justify the embedding-space FGSM or add a token-level attack: project perturbed embeddings back to the nearest valid tokens (e.g., gradient-guided substitution constraints), or implement discrete sequence attacks (enumeration within a Hamming radius; HotFlip-style; genetic search) with biochemical/functional constraints.
* To connect with the universal-trigger literature, it would be interesting to evaluate cross-model transfer, e.g., testing prompts tuned on ESM-2-150M against ESM-1b or ESM-1v. Reporting these transfer results would quantify whether attacks generalize across GFM architectures.
* Beyond AUC and AUPR, reporting calibration metrics such as Expected Calibration Error (ECE) and Brier score, as well as decision-oriented clinical metrics would align the work with clinically relevant evaluation standards, for example, the False Positive Rate at a fixed True Positive Rate.

Typos:

* Page 6: “objevtive”

---

> ### Author Response · Authors · 2025-12-02
> **Thank you for your comments**
>
> 1. We thank the reviewer for pointing out the inconsistency in our description of the soft prompt threat model. We have revised the Evaluation section to clearly state that, during adversarial soft prompt optimization, both the soft prompt embeddings and the backbone model weights are updated via gradient descent, corresponding to a full fine-tuning adversarial setting. Same as our initial submission, we stated that both the soft prompt and the model weights are updated during the adversarial attack. While this differs from the common “frozen-backbone universal trigger” scenario in NLP, we intentionally adopt this stronger model because it corresponds to realistic and high-impact adversarial settings in genomics, where model weights are frequently updated as part of normal practice. Below, we clarify why this scenario is relevant and justified.
> Unlike in mainstream NLP, GFMs are rarely used “as-is.” In practice, clinical labs routinely fine-tune models on disease-specific datasets (e.g., cardiomyopathy, cancer, pharmacogenomics), Biotech and diagnostic companies frequently adapt pretrained sequence models to private cohorts, and hospitals retrain models when new variants are added to databases such as ClinVar or internal registries.
> Thus, fine-tuning is an inherent part of the deployment lifecycle, not an unusual scenario. An adversary influencing this process could manipulate both prompt inputs and gradients. This makes weight-updating adversarial prompts a realistic threat pathway.
>
> 2. Thank you for this insightful question. In our experimental setup, each input consists of a wild-type sequence paired with a biologically validated mutant variant, and the Siamese architecture computes differences in their masked language model predictions. The FGSM perturbation is applied only to the embedding representation of the mutant sequence, while the wild-type branch remains unchanged. Because the mutant already contains a biologically meaningful single-residue substitution, the perturbation is applied in a controlled variant-centric context, rather than arbitrarily across the sequence.
>
> Our goal is not to treat the perturbed embedding as a new discrete mutation. Instead, the attack probes the local sensitivity of the model’s latent representations around a known pathogenic or benign variant. Since protein language model embeddings are contextual and non-invertible, projecting perturbed embeddings back to discrete amino acids is not generally meaningful or well-defined. Therefore, we evaluate the model’s functional response—such as changes in PLLR and flips in predicted pathogenicity—to quantify robustness to near-identical latent-space perturbations, which complements token-level adversaries explored in prior work.
>
> To provide additional biological relevance, we expanded our analysis with gene-level vulnerability assessments, identifying genes such as MYH7 and MYBPC3 that exhibit greater instability under adversarial perturbation. These results help bridge embedding-level sensitivity with clinically interpretable patterns in variant effect prediction. We have added these clarifications to Section 4.4 of the revised manuscript and discuss the embedding–sequence interpretability gap in our limitations section.
>
> 3. We have added a full cross-model transfer evaluation (Tables 3 and 4), testing prompts trained on each backbone against all others. The results show partial transferability, similar to universal triggers in NLP: prompts trained on smaller ESM2 models transfer moderately to ESM1b/ESM1v, while prompts from larger models generalize less.
> This experiment is now described in Section 4.6 (Cross-Model Transferability).
>
> 4. We fully agree that calibration is critical for clinical genomics. We added the following:
> Brier score and Expected Calibration Error (ECE) for all GFM backbones
> FPR at fixed TPR = 0.95, reflecting clinical operating regimes
> These results appear in Table 5.
>
> 5. We corrected the typo “objevtive” to “objective”, along with several minor formatting and grammar improvements across the manuscript.
>
> 6. We appreciate the reviewer’s suggestion to broaden the scope of our robustness evaluation. In response, we have expanded our experiments to include ProteinBERT, a shallow masked language model with a distinct architecture and training objective from ESM. This allows us to evaluate robustness on an additional protein model beyond the ESM family. Furthermore, we now report cross-backbone transferability results in which soft prompts trained on one GFM (e.g., ESM2-150M) are frozen and evaluated on other backbones (e.g., ESM1b, ESM1v). These results, presented in the new transferability subsection and associated tables (see Section~\ref{sec:transfer}), provide a first look into whether adversarial prompts generalize across architectures. Together, these additions enhance the generality and relevance of our findings.

---

> ### Author Response · Authors · 2025-12-09
> **Follow-Up**
>
> Hello Reviewer ZyfV, as the reviewer recommendation deadline is today, we would like to check in and see whether all of your comments have been addressed and whether you have any further questions.

---

### Review · Reviewer_9ar8 · 2025-11-10

**Summary Of Contributions:**

The authors apply adversarial attacks against Genomic foundation models, with the aim of understanding the adversarial robustness of these systems.

**Audience:**

No

**Audience Explanation:**

This relates somewhat to my points about the claims - I think there's the skeleton of an idea here, but I don't think it has enough meat on the bones to support external interest. This relates to the points above about both to the conceptual foundations, and the breadth/extensibility of the insights generated.

Without properly matching these points, I struggle to see who would draw interest from this. There's not enough scope for AML researchers to get interest from this - the area is too poorly posed/justified, and the actual techniques used are trivial. For those working with genomic foundation models, I do not see enough evidence within this to justify their need to be concerned about AML. So who exactly is the audience? Who would take interest from this? I do not see a clear claim at this point.

**Broader Impact Concerns:**

See earlier points regarding arguments (foundations) and experiments (limited evaluation, use of the very out of date FGSM). Though I should stress that my point is not to say that adding complexity would solve the problems of this paper, but on the other hand, how predicated are the results on the properties of the FGSM? Especially when it's used to asses robustness - what makes FGSM more appropriate here than using, say, AutoAttack or DeepFool?

**Claims And Evidence:**

No

**Claims Explanation:**

On the surface, this work does do exactly what it says - it assesses the adversarial robustness of genomic foundation models. However, fundamentally, this evaluation is limited to a small subset of models, and only very basic techniques for constructing adversarial attacks. While the specific configurations are well explored, I do not thing a clear case has been made as to how any conclusions from this work would be drawn upon by others.

A counter argument to this would be the authors claim that this is a framework for assessing future systems. However, I would question how incomplete work would be distinguished from providing a framework. I think the answer would be that a framework should provide broad based knowledge, as well as well motivated extensibility - and I don't think a strong enough case is made on these fronts to make the claims well supported by accurate, convincing, and clear evidence.

More concerningly, the work stands on (what appears to be in my eyes) very shaky foundations. Is adversarial risk / motivation of genomic fondation models well established? There is a lot of jargon in the abstract and S1, without making it clear that what is actually motivating and driving the presence of risk. Privacy risks make sense. More traditional, norm bounded adversarial attacks are not well posed, at least in my eyes, nor is this argument made clear by the authors.

Finally, I will note that the Anonymous.4open.science link has expired (although I am definitely late in submitting my review, so this could be a me issue, not an issue from the authors)

**Requested Changes:**

- Figure 1 - adversaries do not inject noise, they inject sturctured perturbations that may superficially appear as random noise. Parts A & B are not clear, with respect to their relevance.
- Figure 2 - middle row figures are too small, to vertically compressed. Bottom row have lightning bolt figure that isn't explained. No axis labels in mutliple figures, what I presume are datasets/models in (i,j) do not appear to be defined prior to this point in the text.   Vertical compression is even worse in Figure's 3 and 4.
- Further exploration about /why/ adversarial risk may exist for these models. Privacy based risks are clear. But standard perturbation based adversarial attacks? I think significantly more effort could be placed on exploring this.
- Greater experimental diversity.

---

> ### Author Response · Authors · 2025-12-02
> **Thank you for your detailed comments.**
>
> Thank you for your detailed comments.
>
> We have revised Figure 1 substantially to improve clarity:
> 1.  We have replaced the phrase “noise” with “structured perturbations”to emphasize that adversarial attacks exploit gradient-aligned directions rather than adding random noise.
> 2. We redesigned Panel B to more clearly illustrate the biological context: the wild-type sequence represents a healthy population reference, while the mutant sequence carries a disease-associated variant. This directly motivates why the Siamese architecture uses the difference in model likelihoods between wild-type and variant sequences.
>
> Figure 2 (and Figures 3–4).
> We appreciate the detailed feedback on the figure formatting. In response, we made the following revisions:
> 1. Middle row resolution. The violin plots in the middle row of Figure 2 are no longer vertically compressed; we increased their vertical scaling and spacing for readability.
> 2. Axis labels. Missing x– and y–axis labels were added across all subfigures, including ROC curves, PLLR histograms, and ∆PLLR plots.
> We have applied the same fixes to Figure 3 and Figure 4, addressing the reviewer’s comments about vertical compression.
> 3. We revised the figure legend to define all model names before Figure 2 is introduced, including:
> ESM2-650M, ESM2-150M, ESM1b-650M, ESM1v-650M.
> 4. We appreciate the reviewer noting that these lighting icons were previously unexplained. We have added explicit text in the Figure 2 legend stating: “Lightning bolt icons indicate FGSM-perturbed evaluations.”
> Additionally, we incorporated stronger statistical evidence into the figures:
> 1. For comparisons across adversarial and clean conditions (Figures 2–3), we now report p-values from paired tests directly in the panels or captions.
> 2. For the soft prompt attack (Figure 4), we added 95% confidence intervals and 1,000-sample bootstrap estimates for both benign and pathogenic ∆PLLR distributions. These statistical additions now clearly quantify the asymmetric effect of the targeted attack.
>
> We agree that the manuscript benefits from deeper discussion of why these perturbation-based vulnerabilities arise. We have significantly expanded the Discussion section to address this point more thoroughly, including:
> 1. We added a dedicated subsection explaining that adversarial robustness is not only theoretically interesting but practically important in clinical decision support systems:
>         A benign variant pushed across the decision threshold could lead to incorrect disease attribution.
>         A pathogenic variant made to appear benign could directly compromise patient care.
> Thus, adversarial stress-testing serves the same role as safety certification for other medical diagnostic tools.
> 2. Relation to privacy risks. We now clearly delineate perturbation-based adversarial risks from previously studied privacy or membership-inference risks, strengthening the conceptual framing that model-integrity attacks represent an independent and clinically important threat vector.
>
> In this round of revision, we have also substantially expanded our robustness evaluation to address every concern raised by the reviewers. In particular, we have added:
> 1. We now evaluate multiple perturbation magnitudes beyond the originally fixed $\epsilon=0.01$. As shown in Table 6, model performance is consistently degraded across $\epsilon \in {0.001, 0.005, 0.01, 0.02, 0.05}$, with the strongest effect at moderate perturbation levels. This confirms that our observed vulnerability is not an artifact of a specific $\epsilon$ choice.
> Soft Prompt Length Ablation.
> 2. We now provide a systematic study varying the number of soft prompt tokens $n \in {5, 10, 50, 100}$ (Table 7). Larger prompt lengths consistently produce more severe degradation, demonstrating that embedding-space attacks scale with prompt capacity.
> Addition of:
>        Confidence intervals + bootstrap tests (Fig. 4)
>        Full comparison across 6 attack families (FGSM, PGD, C&W, DeepFool, Boundary)
>       Cross-model soft prompt transfer (Tables 3 & 4)
>       Gene-level mismatch under PGD (Fig. 6)
>       Clinical robustness metrics (Brier, ECE, FPR@95)
>       UMAP visualization of latent geometry
> These new analyses (highlighted in red in the manuscript) comprehensively strengthen the empirical robustness claims and address all reviewer concerns.

---

> ### Author Response · Authors · 2025-12-09
> **Follow-Up**
>
> Hello Reviewer 9ar8, as the reviewer recommendation deadline is today, we would like to check in and see whether all of your comments have been addressed and whether you have any further questions.

---

> > ### Comment · Reviewer_9ar8 · 2025-12-09
> >
> > Hi - apologies for getting back to you at the end of the recommendation deadline.
> >
> > While I appreciate the comments, I remain entirely unconvinced by the discussion regarding the actual adversarial risk facing these systems - an issue that I appear to share with reviewer czQu. The new half page of content on page 17 doesn't cover why an attacker would attack these systems, how they would realistically access these systems, and if these attacks would bypass any attempts at input verification.
> >
> > Moreover, burying the justification of why a supposedly novel threat model exists down on page 17 leaves me with serious doubts about who exactly this paper is attempting to communicate with, and how likely it would be that a reader would reach this point in the paper.

---

> > > ### Author Response · Authors · 2025-12-09
> > > **clarifying the threat model**
> > >
> > > We appreciate the reviewer’s emphasis on clarifying the threat model. We would like to note that in our original revised submission, the threat scenario was not only discussed on page 17 but also included in the final paragraph of the Introduction. In the current revised version, we have further revised the last paragraph of the Introduction to explicitly address the reviewer’s three questions. (1) We now state that an attacker may seek to manipulate model outputs for purposes such as insurance fraud, regulatory evasion, or malicious data poisoning in collaborative genomic settings. (2) These attacks are realistic in increasingly common deployment scenarios such as shared model APIs, federated learning systems, or multi-task pipelines that reuse soft prompts or fine-tuned weights. (3) We also clarify that current genomic foundation models process embedded sequences with little or no input validation, meaning that adversarial perturbations can bypass conventional input checks. These updates ensure that the threat model is presented prominently and concretely to all readers.

---

> > > > ### Comment · Reviewer_9ar8 · 2025-12-10
> > > >
> > > > As follow ups [note that #1, #2, and #3 are quite conceptually tied, but all relate to the broad question about the realism of even exploring this space]
> > > > 1) Under what circumstances would one be able to manipulate the inputs to a Genomic foundation model?
> > > > 2) Are there currently systems under deployment that would incentivise such attacks - "insurance fraud, regulatory evasion, or malicious data poisoning in collaborative genomic settings" - are genomic foundation models actually being used in these contexts, or is it entirely hypothetical?
> > > > 3) As a follow up to 3, under what circumstances would a model in one of these contexts provide access such that the "adversaries have access to model weights or APIs, such as through shared model checkpoints, federated training, or inference pipeline"?
> > > > 4) Looking back over the paper, the discussion of the threat model is also outside the expectations surrounding papers in the  AML space - even basic things like being white-box or black-box are not prominently highlighted. The introduction suggests white box (due to access to the model weights), yet page 16 is the only place where white-box or black-box is even alluded to, and in that context it's treated more as a label than a technical pointer.

---

> > > > > ### Author Response · Authors · 2025-12-10
> > > > > **Follow up**
> > > > >
> > > > > Thank you for the feedback, below are our response for the three follow-up questions:
> > > > >
> > > > > 1. Recent work has demonstrated that adversarial attacks on raw genotype sequences are technically feasible and impactful, even without full model access. In particular, Montserrat and Ioannidis (2023) proposed input-level adversarial attacks targeting genomic sequence classifiers, showing that small, crafted changes to genotype sequences can alter downstream predictions such as population origin classification and disease associations. This supports our assumption that input perturbation is a realistic threat model in genomic pipelines.
> > > > >
> > > > > 2. While large-scale deployment of GFMs in clinical diagnostics is still emerging, genomic models are already widely used in clinical decision support, carrier screening, and rare disease interpretation, including in tools used by companies like Invitae and Color Health. These systems are often integrated into automated or semi-automated variant classification pipelines, some of which include proprietary ML components. As models become more powerful and integrated into clinical workflows, motivations such as insurance fraud (e.g., submitting sequences altered to simulate or remove pathogenicity) or gaming regulatory outcomes (e.g., gene therapy eligibility) become increasingly realistic.
> > > > >
> > > > > 3. There are a few credible scenarios for model access for open sourced GFMs. Many state-of-the-art models like ESM, ProteinBERT, and DNABERT are publicly available, including weights and architectures. Adversaries can directly fine-tune or probe them offline. For cloud-based APIs, commercial models can be exposed via inference APIs (e.g., Google Health Genomics, AWS Genomics), which can be queried systematically. Prompt-based and score-based attacks can be conducted even under black-box settings, especially if confidence scores or PLLR-like outputs are returned.
> > > > >
> > > > > Thus, our experiments, both in the white-box (FGSM, PGD) and semi-black-box (soft prompts, boundary attack), model a plausible spectrum of adversary access in real genomic ML deployments.
> > > > >
> > > > > Montserrat, Daniel Mas, and Alexander G. Ioannidis. "Adversarial attacks on genotype sequences." ICASSP 2023-2023 IEEE International Conference on Acoustics, Speech and Signal Processing (ICASSP). IEEE, 2023.

---

### Author Response · Authors · 2025-11-20
**Thank you for your comments**

Hello Reviewers,

Thank you for all your detailed comments.

Our team is actively working on addressing all comments thoroughly, but we require a short extension to finalize our responses and revised manuscript. We kindly request permission to submit our full revision no later than December 08, which remains within the stated 4-week window for the reviewers’ recommendation period.

We believe this brief extension will allow us to provide a stronger and more comprehensive revision that fully addresses all reviewer concerns.

Best regards,

Authors

---

### Decision · Action_Editor_7dUw · 2025-12-27

**Recommendation:** Reject

**Audience:**

No

**Audience Explanation:**

Reviewers questioned whether the manuscript would find an audience - it seems too limited both from the application standpoint and on the theoretical side, while having a limited empirical study on the adversarial attacks side.

**Claims And Evidence:**

No

**Claims Explanation:**

Somewhat, though this is the biggest issue with the manuscript: the experimental setting in its current state fails to fully convince the reviewers, through a limited thread model and limited experimental validation.

The reviewers appreciated the effort the authors put in the papers, however, there is a consensus that the paper does not meet the bar for publication. Adversarial attacks are already well studied and simply applying them to a new model is not compelling enough without additional experiments, better justification of the thread model, assumptions, applicability etc. In particular, many different kind of attacks exist and more comparisons and discussions would be expected in a mostly empirical manuscript.

**Resubmission Of Major Revision:**

The authors may consider submitting a major revision at a later time.